# Seabird colonies as important global drivers in the nitrogen and phosphorus cycles

Xosé Luis Otero [1], Saul De La Peña-Lastra [1], Augusto Pérez-Alberti[2], Tiago Osorio Ferreira [3] & Miguel Angel Huerta-Diaz [4]

Seabirds drastically transform the environmental conditions of the sites where they establish their breeding colonies via soil, sediment, and water eutrophication (hereafter termed ornitheutrophication). Here, we report worldwide amounts of total nitrogen (N) and total phosphorus (P) excreted by seabirds using an inventory of global seabird populations applied to a bioenergetics model. We estimate these fluxes to be 591 Gg N y$^{-1}$ and 99 Gg P y$^{-1}$, respectively, with the Antarctic and Southern coasts receiving the highest N and P inputs. We show that these inputs are of similar magnitude to others considered in global N and P cycles, with concentrations per unit of surface area in seabird colonies among the highest measured on the Earth's surface. Finally, an important fraction of the total excreted N (72.5 Gg y$^{-1}$) and P (21.8 Gg y$^{-1}$) can be readily solubilized, increasing their short-term bioavailability in continental and coastal waters located near the seabird colonies.

[1] Departamento de Edafoloxía e Química Agrícola, Campus Vida, Facultade de Bioloxía, Universidade de Santiago de Compostela, 15782 Santiago de Compostela, Spain. [2] Departamento de Xeografía Física, Facultade de Xeografía e Historia, Universidade de Santiago de Compostela, 15782 Santiago de Compostela, Spain. [3] Luiz de Queiroz College of Agriculture, University of Sao Paulo (ESALQ-USP), Av. Pádua Dias 11, CEP 13418-900 Sao Paulo, Brazil. [4] Instituto de Investigaciones Oceanológicas, Universidad Autónoma de Baja California, Carretera Transpeninsular Ensenada-Tijuana No. 3917, Fraccionamiento Playitas, CP 22860 Ensenada, Baja California, Mexico. Correspondence and requests for materials should be addressed to M.A.H.-D. (email: huertam@uabc.edu.mx)

Worldwide, seabirds act as biological pumps between marine and terrestrial ecosystems[1,2]. As a result of the gregarious nature of seabirds, extremely high densities of birds can be reached in breeding colonies, leading to the accumulation of large amounts of debris in coastal ecosystems. For example, fecal material in penguin colonies from Marion Island represented about 85% of all organic debris deposited on the substrate, with huge amounts of (~100 Mg dry weight) accumulating in the colonies during the nesting season[3]. This fecal material, known as "guano" (a term derived from the Quechuan word for dung or animal excrement), contains high concentrations of macro and micronutrients[4–6], and has been used since ancient times as a natural fertilizer[7,8].

In addition to the eco-historical importance of guano, many researchers have investigated the effect that seabird colonies have on the biogeochemical processes and vegetation ecology at different geographical scales (local and regional, Fig. 1)[4,6,9–13]. The accumulation of organic matter and nutrients have caused important environmental changes in coastal ecosystems[4,9,10]. These changes include physical disturbances (treading, collection of nest materials), chemical changes in soil composition (guano and salt deposition), and alteration of competitive processes

(dispersal of allochthonous seeds, expansion of annual or ruderal species)[9,10]. In 1936, Vevers[11] carried out one of the first studies of these impacts and, on studying the vegetation of the island of Ailsa Craig (Scotland), concluded that seabird colonies were one of the main factors influencing the flora where colonies were established. A few years later, in a study carried out in Jan Mayen Island in the Arctic region, Russell et al.[12] reported that in 1940 the amount of nitrogen derived from dead plant tissues was small due to the low temperatures in the region, which restricted the activity of the microorganisms responsible for decomposing the organic matter. However, the nitrogen supplied by seabird colonies would exert an important effect on the development of Arctic plant communities and the appearance of new plant taxa.

Although in some cases seabird colonies have profoundly altered the biogeochemical processes that occur in coastal surface systems (soils, sediments, and waters), and have transformed plant communities (for example, Mediterranean and Atlantic islands[6,9], North-East Scottish coast[11], and Pacific reef corals[13]), most studies revealing biogeochemical and ecological alterations have been mostly of local interest and importance to particular areas[6,9–14] (Fig. 1). Recent studies have attempted to show that seabird colonies may have regional or global effects on the cycling

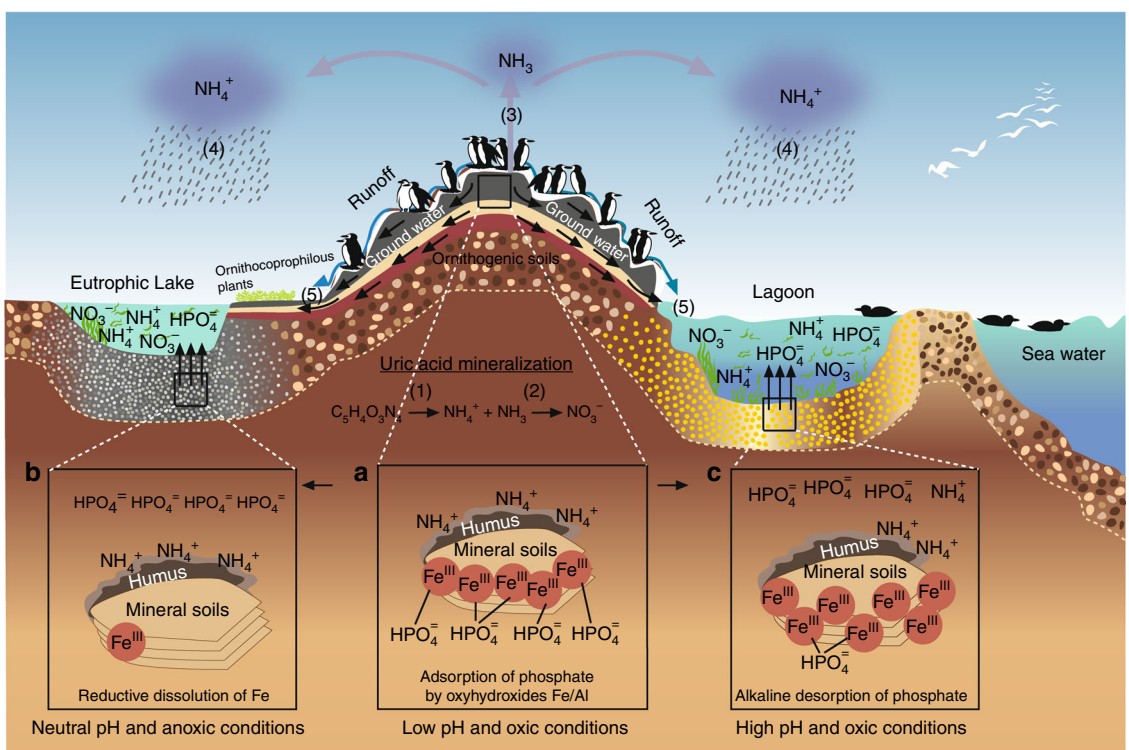

**Fig. 1** Seabird ornitheutrophication coupling. Schematic summary of processes coupling local and regional environmental effects in seabird colonies. Colonies can be considered as nutrient hot spots, especially, for N and P. Nitrogen is the key nutrient in marine environments and phosphorous in continental waters. Both are found in high concentrations in seabird feces. Uric acid is the dominant N compound, and during its mineralization different N forms are produced: (1) ammonification produces $NH_3$ and $NH_4^+$, and (2) nitrification produces $NO_3^-$ by $NH_4^+$ oxidation. Under the alkaline conditions, typical of the seabird feces, the $NH_3$ is rapidly volatized (3) and transformed to $NH_4^+$, which is transported out of the colony, and through wet-deposition exported to distant ecosystems, which are eutrophized (4). Similarly, nutrients in the colony can be leached and transported out through runoff and groundwater seepage (5), generating in cases (4) and (5) environmental impacts at the regional level. On the other hand, $NH_4^+$ in soil (ornithogenic soils) can be adsorbed by organominerals and remain as an exchangeable cation (panel a). The soil $NH_4^+$ in the colony can be oxidized to nitrate through nitrification processes, and rapidly washed to subterranean or superficial waters, eutrophizing nearby ecosystems (local impact, 5). In both cases, the $NO_3^-$ and $NH_4^+$ can reach creeks and small lakes, eutrophizing them (regional impact). Phosphorus cycle is simpler and has a rather reduced mobility. This element is found in a number of chemical forms in the seabird fecal material, but the most mobile and bioavailable is orthophosphate ($HPO_4^=$), which can be lixiviated (solubilized) to subterranean or superficial waters (5). However, an important fraction of the P can be adsorbed by Fe/Al oxyhydroxides in acidic soils. Through erosion, these colloids can reach anoxic freshwater or estuarine sediments, where P is liberated to the water column by the reductive dissolution of Fe(III) oxyhydroxides (panel b). If the colloids reach oxic marine sediments, P can still be liberated to the water by alkaline desorption (panel c), a process that involves changes in the surface charge of Fe/Al oxyhydroxides. In both cases, water eutrophization is produced

| Table 1 Estimated total and labile forms of excreted N and P arranged by seabird order | | | | | | |
| --- | --- | --- | --- | --- | --- | --- |
| Order | Breeding birds and chicks (millions) | Number of species | Total N (Gg N y⁻¹) | Labile N (Gg N y⁻¹) | Total P (Gg P y⁻¹) | Labile P (Gg P y⁻¹) |
| Charadriiformes | 291 | 127 | 116 | 9.8 | 19 | 5.1 |
| Pelecaniformes | 30 | 53 | 51 | 6.5 | 9 | 1.5 |
| Procellariiformes | 424 | 123 | 117 | 21.2 | 20 | 4.3 |
| Sphenisciformes | 59 | 17 | 307 | 35.0 | 51 | 10.9 |
| TOTAL | 804 | 320 | 591 | 72.5 | 99 | 21.8 |

of elements such as N, which is present at high concentrations in the fecal materials of seabirds (total N ~1–25%; e.g., refs. [15–17]). In the last decade, particular attention has been given to the atmospheric emission of ammonia ($NH_3$) via the mineralization of uric acid present in seabird excrements. In a study estimating global $NH_3$ emissions due to seabird colonies, Riddick et al.[18] concluded that these colonies are the main source of $NH_3$ emission to the atmosphere in remote areas, and that these emissions may significantly affect ecosystems, both within and outside the colonies (Fig. 1).

Phosphorus also limits primary productivity in both terrestrial and aquatic environments[19,20] and is also found in high concentrations in seabird excrement (total P: 0.09–17%;[6]). However, global inputs of P from seabird colonies have not yet been estimated. In the present study, the worldwide amounts of these two elements that are excreted by seabirds and their chicks were estimated in breeding colonies. For this purpose, current estimates of the world seabird populations were obtained from global seabird population data published by international organizations. A bioenergetics model (proposed by Wilson et al.[21] and later used by Riddick et al.[18]) was used to calculate the amounts of N, and then adapted to calculate the quantities of P excreted by reproducing seabirds and their chicks in colonies worldwide. Finally, the importance of seabirds within the global context of N and P cycling is discussed by considering the main intercompartmental flows of these two elements.

Results obtained in this work indicate that N and P excreted by seabirds in their colonies are similar in magnitude to the values of other fluxes that are normally included in the global biogeochemical cycles of these two elements. Hence, it is proposed that a mass transfer of N and P from the marine environment to seabird colonies should also be taken into account when balancing their fluxes at the global scale.

## Results

**Seabird population estimates.** The worldwide population of breeding seabirds and chicks is estimated to be 804 million individuals (Table 1 and Supplementary Data 1), and the total population, including 30% of non-breeding seabirds is estimated to be 1045 million individuals. Similar results have been obtained in previous studies, with total population estimates ranging from 900 to 1180 million[18,22].

Of the total number of seabird marine species, 6% present populations above 10 million individuals (Supplementary Data 1), with the most numerous species being: Antarctic Prion (*Pachyptila desolata*, 50 million), Little Auk (*Alle alle*, 26 million), Least Auklet (*Aethia pusilla*, 24 million), Short-tailed Shearwater (*Ardenna tenuirostris*, 23 million), Northern Fulmar (*Fulmarus glacialis*, 22.5 million), and Thick-billed Murre (*Uria lomvia*, 22 million). By far the largest order is Procellariiformes, with 424 million individuals and 123 species, followed by Charadriiformes with 291 million and 127 species (Table 1). Global distribution of the seabird colonies shows that they are distributed mainly in the polar zones (Fig. 2; Supplementary Data 2), with more than half of the total population concentrated in Antarctica and its sub-

Antarctic islands (213 million) and in Greenland and Svalbard islands (209 million). However, despite a similar distribution in the total number of seabirds between polar zones, it should be taken into consideration that large population sizes do not necessarily correspond to large nutrient excretions[18]. The differences between species' body masses and length of the breeding seasons are the main reasons why nutrient excretions in Antarctica and its sub-Antarctic islands are far larger than the ones obtained for Greenland and Svalbard islands. For example, species from the Arctic zone are small in size and weight, with the body masses of the two most abundant species (Little Auk and Least Auklet) in the order of 0.15–0.18 kg[23] and ~0.08 kg[24], respectively. However, an important portion of the species present in Antarctica and its sub-Antarctic islands are big in size and weight, as is the case with the Chinstrap (*Pygoscelis antarcticus*, 3–5 kg[25]) and Emperor (*Aptenodytes forsteri*: 22–37 kg[26]) penguins. These differences in body mass have a dramatic effect on the quantity of excreted N and P, as discussed in the following section.

**Global N and P excretion in breeding colonies.** Worldwide total N and P excreted in breeding colonies are estimated to be 591 and 99 Gg y⁻¹, respectively (Table 1). The amounts of N and P mobilized by seabirds are 6.4 times higher when the total population of breeding and non-breeding birds are taken into account, and the breeding and non-breeding seasons are considered in the calculation (3800 Gg N y⁻¹, 631 Gg P y⁻¹). Nonetheless, outside of the breeding season, most seabirds disperse with, consequently, small effects on nutrient concentrations in coastal areas.

The nesting populations in the Antarctic and Southern Ocean regions account for 80% of the total N and P excreted (470 Gg N y⁻¹, 79 Gg P y⁻¹; Table 2); however, the overall seabird population (breeding and chicks) of this region represents only 26% of the global population (213 million individuals). The second largest input corresponds to Greenland and the Svalbard islands, which although they are home to a similar-sized population (26% of the total), receive 14 times less N and P than the Antarctic region. Australasia occupies the third position, with values similar to the other regions, but with a smaller population (12% of the total). The colonies at mid-latitudes (i.e., Atlantic, Middle East, Europe, Asia, etc.) contribute with relatively modest amounts of N and P to the total amounts excreted, although these colonies represent a high proportion of the global population of seabirds (Fig. 2; Table 2).

By order, the Sphenisciformes show the major global contribution, with 307 Gg N y⁻¹, 51 Gg P y⁻¹ (Table 1), with the most important species being the Macaroni (*Eudyptes chrysolophus*, 108 Gg N y⁻¹, 18 Gg P y⁻¹) and King (*Aptenodytes patagonicus*, 60 Gg N y⁻¹, 10 Gg P y⁻¹) penguins (Fig. 3). These two species live in the sub-Antarctic and breed in many of the sub-Antarctic islands located between 46 and 55 degrees south, including Southern Chile, Falkland Islands, South Georgia, and Islands of South Africa and South Australia. In these places the colonies of Macaroni and King Penguins comprise more than one million and half a million individuals, respectively. The second

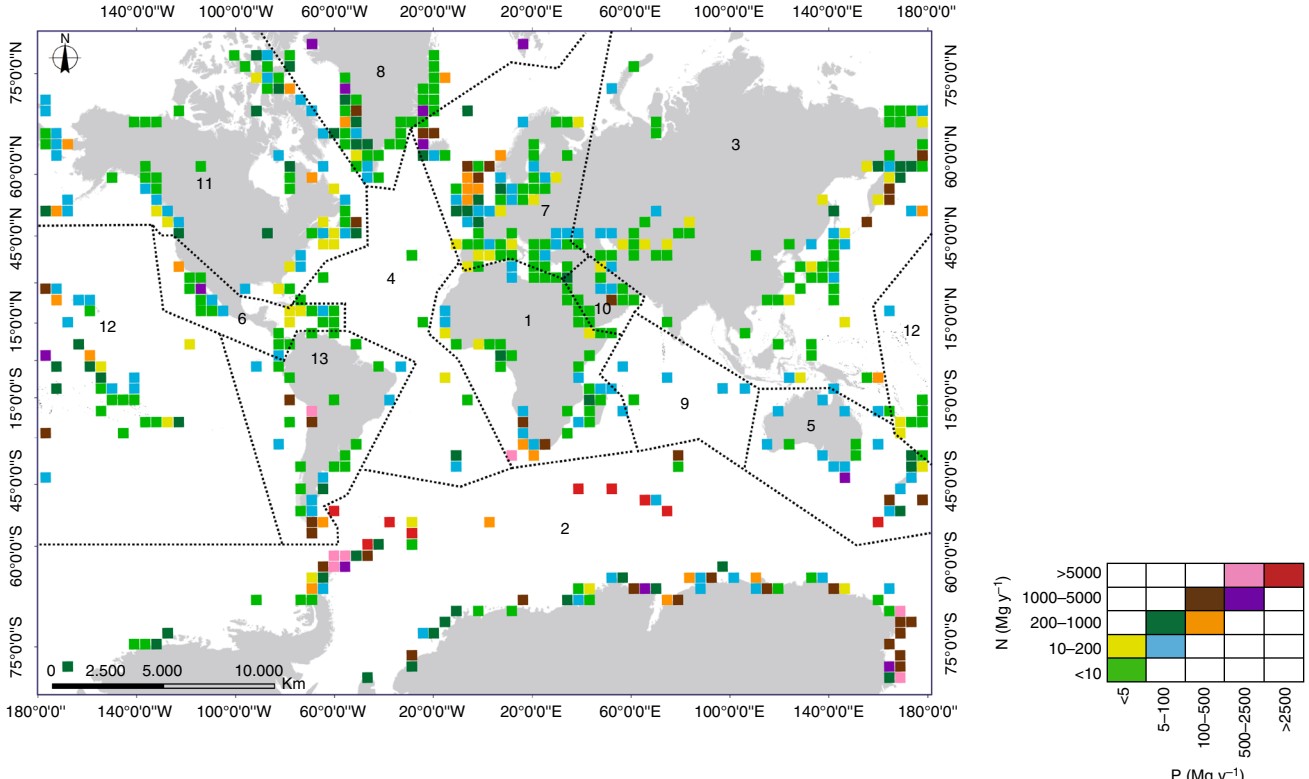

**Fig. 2** Global distribution of N and P excretion by seabird colonies. To show the colony distributions in a clear way, a fishnet grid of square cells with 500 km sides and the sum of the number of seabirds included in each cell was generated. Lines delineate regional boundaries after Riddick et al.[18]: 1. Africa, 2. Antarctica and Southern Ocean, 3. Asia, 4. Atlantic, 5. Australasia, 6. Caribbean and Central America, 7. Europe, 8. Greenland and Svalbard, 9. Indian Ocean, 10. Middle East, 11. North America, 12. Pacific and 13. South America. Scale shows fluxes in Mg y$^{-1}$, where 1 Mg = 1 × 10$^6$ g

### Table 2 Biogeographic distribution of total N and P excreted by seabird breeders and chicks

| Region | Total population of breeders and chicks (millions) | Percentage of total seabirds | N excreted (Gg N y$^{-1}$) | Percentage of total N excreted | P excreted (Gg P y$^{-1}$) | Percentage of total P excreted |
|---|---|---|---|---|---|---|
| Antarctica and Southern Ocean | 213 | 26 | 470 | 80 | 79 | 80 |
| Greenland and Svalbard | 209 | 26 | 32 | 5.5 | 5.4 | 5.5 |
| North America | 73.9 | 9.2 | 7.7 | 1.3 | 1.3 | 1.3 |
| Australasia | 95.5 | 12 | 27 | 4.5 | 4.5 | 4.6 |
| Pacifica | 106 | 13 | 12 | 2.0 | 1.9 | 2.0 |
| Europe | 30.5 | 3.8 | 8.2 | 1.4 | 1.4 | 1.4 |
| South America | 12.6 | 1.6 | 11 | 1.9 | 1.8 | 1.9 |
| Asia | 40.1 | 5.0 | 8.2 | 1.4 | 1.4 | 1.4 |
| Indian Ocean | 12.3 | 1.5 | 2.7 | 0.46 | 0.46 | 0.46 |
| Atlantica | 0.31 | 0.04 | 1.1 | 0.19 | 0.18 | 0.18 |
| Middle East | 1.23 | 0.15 | 1.6 | 0.28 | 0.28 | 0.28 |
| Africa | 6.16 | 0.77 | 5.5 | 0.93 | 0.92 | 0.93 |
| Caribbean and Central America | 3.08 | 0.38 | 3.8 | 0.65 | 0.64 | 0.65 |

major contribution corresponds to the order Procellariiformes, which produce 117 Gg N y$^{-1}$, 20 Gg P y$^{-1}$, with the Northern Fulmar (*Fulmarus glacialis*, 27 Gg N y$^{-1}$, 4 Gg P y$^{-1}$) and the Short-tailed Shearwater (*Ardenna tenuirostris*, 22 Gg N y$^{-1}$, 4 Gg P y$^{-1}$) contributing the most to N and P deposition. The Northern Fulmar breeds throughout the North Atlantic and North Pacific, with the largest populations in Alaska and Korea, whereas the Short-tailed Shearwater breeds in Tasmania and off the coast of South Australia (Supplementary Data 1).

Similar values are obtained for the order Charadriiformes, in which the two species contributing most to the deposition of N and P are the Common Guillemot (*Uria aalge*, 21.6 Gg N y$^{-1}$, 3.6 Gg P y$^{-1}$) and the Thick-billed Murre (*Uria lomvia*, 21 Gg N y$^{-1}$, 3.5 Gg P y$^{-1}$). These two species have a circumpolar distribution in the Arctic and in the high Arctic regions of North America, Europe, and Asia. The most numerous populations of the Common Guillemot are found in Canada and those of the Thick-billed Murre in Alaska, Northern Canada, and Southwest

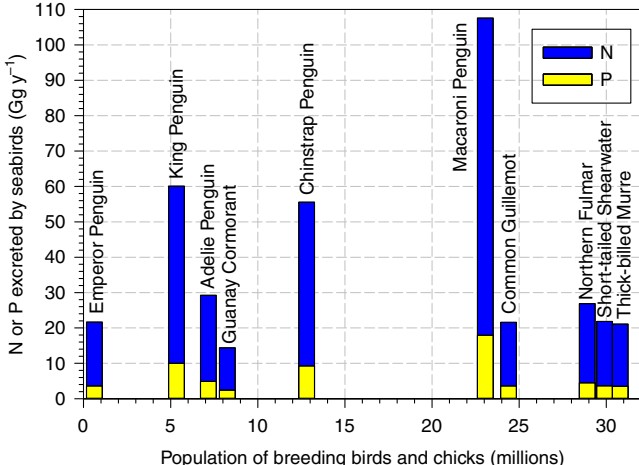

**Fig. 3** The ten seabird species excreting the most N and P on a global scale relative to population size. The relevance of seabirds species in terms of N and P fluxes from marine to continental environments (breeding colonies) depends on a number of factors, such as population size, corporal weight, length of the breeding season, or type of feeding and, hence, their contribution is very uneven. Penguins are the main contributors, fundamentally because of their high body mass (height: 70–130 cm; weight: 5–40 kg) and the long period of time they remained in the colony (more than one year), whereas the contribution of smaller species such as the Common Gillemot, Northern Fulmar, Short-tailed Shearwater, or Thick-billed Murre is a consequence of their large population sizes

Greenland, where colonies of more than one million individuals occur. The lowest N and P depositions are by members of the order Pelecaniformes, whose main species were the Guanay Cormorant (*Phalacrocorax bougainvilliorum*, 14 Gg N y$^{-1}$, 2.4 Gg P y$^{-1}$), and the Great Cormorant (*Phalacrocorax carbo*, 6.7 Gg N y$^{-1}$ and 1.1 Gg P y$^{-1}$). The Guanay Cormorant includes approximately 8 million breeders and chicks in diverse colonies, which is found mainly along the Pacific coast of Peru and Northern Chile. The Great Cormorant is widely distributed, being found on every continent, except South America and Antarctica. The largest colonies are found in Estonia (Europe, ~10.000 birds) and Mauritania (Africa, ~14.000 birds).

Ten species contribute more than 60% of the total N and P excreted (Fig. 3), partly due to their large populations (e.g., the Macaroni Penguin). However, it is interesting to notice that large populations do not necessarily excrete the greatest amounts of N and P (Fig. 3), as the amount of fecal material excreted depends on the size of each bird and its residence time in the colony. The species that excrete the largest amounts of N and P per individual include five species of penguins and four of albatrosses, representing the biggest birds (Supplementary Fig. 1A). The Emperor Penguin represents a large contributor, because of its big size and its long residence time in the colonies (~330 days)[26].

Chicks generally deposit less N and P than adults, usually not more than 4% of the contribution made by the adult birds (Supplementary Fig. 1). This is mainly due to the importance of reproductive success, in addition to body mass and residence time in the colony. Hence, the species that produce the most N and P include four species of cormorants, as well as penguins, and albatrosses. The chicks of the Guanay Cormorant and Great Cormorant are the ones that excrete the most N and P. They contribute 21 and 26%, respectively, of the total excreted by these species, presumably due to their large size, long residence time in the colonies, and high reproductive success (2.4 and 2.16 chick pairs y$^{-1}$, respectively). On the other hand, the amounts of excreted labile forms of N and P (those that can be readily

dissolved) are 72.5 Gg y$^{-1}$ and 21.8 Gg y$^{-1}$, respectively, with the highest values corresponding to the Sphenisciformes (35.0 Gg N y$^{-1}$ and 10.9 Gg P y$^{-1}$, respectively), followed by the Charadriiformes, Procellariiformes, and Pelecaniformes (Table 1).

## Discussion

Nitrogen and phosphorus are both considered as essential elements for all forms of life, and the corresponding biogeochemical cycles of these two elements are two of the most important in the biosphere, since these elements limit primary production in marine and terrestrial environments[19,27–29]. However, despite their importance in biogeochemical processes, some of the paths comprising the global cycles of N and P are not completely known[30,31]. The N cycle is considered as a "perfect" cycle, in which the reservoirs are easily accessible and includes numerous feedback controls. By contrast, the P cycle is termed "imperfect", as turnover of the buried sedimentary reservoir is determined by tectonic events acting over tens and hundreds of million years[32]. Nonetheless, although both elements are relatively abundant in the Earth's surface (Supplementary Tables 1, 2), most N and P fractions are not directly available to organisms. The main nitrogen compartment is atmospheric $N_2$ (Supplementary Table 1), which is almost inert and can only be transformed into bioavailable forms (e.g., $NH_3$) via highly energetic processes (e.g., lightning, the Haber–Bosch process, or biological fixation of N;[30]). Moreover, the P present in bedrock, soils, and sediments is not directly available to organisms[30].

Taking the above into account, various authors have suggested that seabird colonies represent a positive geochemical anomaly (i.e., above background values) regarding the concentrations of N and P present in soils, sediments, and water[1,6,13,33]; hereafter, termed ornitheutrophication (Fig. 1). However, these authors have only considered the results of studies carried out at local scales and, as far as it is known, only one study has considered the global importance of seabirds. Riddick et al.[18] extrapolated the impact of atmospheric emissions of $NH_3$ from the mineralization of uric acid present in seabird excrements. These and other authors reported that global emissions from seabird excrements may range between 97 and 442 Gg $NH_3$ y$^{-1}$, making them an environmentally relevant process[33–35].

Most compartments of the global N biogeochemical cycle contain between three and six times more N than that excreted by seabirds in the breeding colonies (Supplementary Table 1). However, the magnitude of the flows between marine and terrestrial environments by breeding seabirds ($0.59 \times 10^3$ Gg N y$^{-1}$) and by the total seabird population ($3.8 \times 10^3$ Gg N y$^{-1}$) are of the same order of magnitude as that mobilized via other processes, such as lightning ($5.0 \times 10^3$ Gg N y$^{-1}$), N fixation by rice cultivation ($5.0 \times 10^3$ Gg N y$^{-1}$), inputs to the ocean via groundwater ($4.0 \times 10^3$ Gg N y$^{-1}$), and sea-to-land N transfer via commercial fisheries ($3.7 \times 10^3$ Gg N y$^{-1}$). Similar results were obtained for P, with its flow from marine to terrestrial environments attributed to seabirds (breeding seabirds: $0.10 \times 10^3$ Gg P y$^{-1}$; total population: $0.63 \times 10^3$ Gg P y$^{-1}$) being of similar magnitude than those occurring between oceanic waters and atmosphere ($0.31 \times 10^3$ Gg P y$^{-1}$), those produced by fishing activities ($0.32 \times 10^3$ Gg P y$^{-1}$) or those attributed to the dissolved inorganic P flux of rivers ($0.8$–$1.4 \times 10^3$ Gg P y$^{-1}$) (Supplementary Table 2). Results indicate that mass transfer calculations of N and P from marine to terrestrial environments should also be taken into account when balancing the fluxes of the global biogeochemical cycles of these two elements. Furthermore, the mobilization and concentration of N and P by seabirds is found to be particularly important when the inputs per unit of surface area are taken into account. Thus, for a Macaroni Penguin colony these inputs can be as high as[18]

114,240 kg N ha$^{-1}$ y$^{-1}$, whereas those for a Northern Gannet colony can reach[33] 52,200 kg N ha$^{-1}$ y$^{-1}$. These values are highest known for the Earth's surface, representing between 500 and >1100 times the total annual inputs of N from agriculture[36,37] (~65–80 kg N ha$^{-1}$).

It is important to mention that unlike other major compartments, a large proportion of the excreted N and P is present in highly bioavailable forms. The N found in seabird excrements occurs mainly as uric acid (~80%) and, to a lesser extent, as ammonia and proteins[15]. Uric acid is rapidly mineralized to highly bioavailable inorganic forms, such as NH$_3$ and NO$_3^-$ (Fig. 1, Supplementary Table 3). Riddick et al.[18,38] estimated that volatization of the N contained in seabirds fecal material to form NH$_3$ was in the range 2–5% in colonies located in cold climates, whereas in tropical zones, this range increased to 31–65%. High mass values were reported by Lindeboom[15] for the Marion Island ecosystem, who suggested that the NH$_3$ associated with fecal matter was either blown out to the sea (184 kg NH$_3$ day$^{-1}$), or deposited on the island (30 kg NH$_3$ day$^{-1}$). The subsequent N deposition on the island creates an NH$_3$ shadow in the nearby vegetation, where species characteristic of coastal areas often disappear (e.g., *Armeria maritima, Umbilicaria decussata,* and *Usnea sphacelata*;[39–41]), whereas other species associated with bird life (ornithocoprophilous species such as *Cochlearia groenlandica, Saxifraga rivularis, Festuca vivipara, Poa cookii,* and *Callitriche antarctica*), grow more vigorously than in other parts of the island[15,39,40]. In addition to ammonia emission to the atmosphere, N can also be mobilized to coastal waters or lakes by dissolution in runoff waters, thus increasing the rate of primary production of plant life in coastal ecosystems[41]. Results show that 12.7% (0.07 × 10$^3$ Mg y$^{-1}$) of the total excreted N corresponds to labile forms of this element, which can be readily lixiviated toward continental or coastal waters (Table 1). This process can become especially relevant in the Antarctic and Southern Ocean regions, where the majority of the main guano producers, represented by the Sphenisciform population is concentrated (Table 1).

In regard to labile P, one of its main differences relative to N is that the former lacks a stable gaseous phase (phosphine or PH$_3$)[42], making the concentrations of this compound in seabird excrements extremely low[43]. Results show that the global emission of P to the atmosphere from fecal material produced by seabirds can be considered as negligible (1.5 × 10$^{-6}$ Gg y$^{-1}$). On the other hand, mobilization of P via leaching can be restricted due to its adsorption to soil colloids[6] (Fig. 1). Thus, although annual losses of N may occur via leaching[44,45], P accumulates in the soil, with extremely high values maintained over time[46]. Otero et al.[6] found a clear accumulation of P in soils of Yellow-legged Gull (*Larus michahellis*) colonies located in the National Park of Atlantic Island (Galicia–NW Spain). The total P concentrations in soils under a colony of Yellow-legged Gulls were, on average, three times higher than in an area without seabirds, and the P available to plants was 30 times higher. However, the majority of the seabird colonies' soils are located on rocky substrates, with shallow and sandy soils or where the occurrence of permafrost near the soil surface has a strong regulatory effect on leaching and development processes[47,48]. Under these conditions, a rapid P soil saturation is eventually reached, and this element is then lixiviated toward coastal or continental waters[6]. In this sense, the results obtained in this work show that 21% (0.02 × 10$^3$ Gg y$^{-1}$) of the total excreted P is easily lixiviated. This labile phosphorus flux can be considered as important, since it represents 2.5 and 10% of the total dissolved fluvial fluxes of inorganic and organic P, respectively (Supplementary Table 3).

The enrichment of labile P in seabird colony soils has recently been recognized as important to the extent that the world

reference base for soil resources[47] has included ornithogenic material as a diagnostic material for soil classification. Ornithogenic material is characterized by being strongly influenced by bird excrement and by high concentrations of P soluble in 1% citric acid. Recent studies carried out in penguin colonies in the Antarctic refer to the process of phosphatization as a new pedogenetic process, leading to the formation of phosphate minerals, such as taranakite, minyulite, leucophosphite, and struvite[48,49]. The origins of these minerals are known to be related to the fecal material produced by seabirds during long periods of time[50]. Phosphates occurring in ornithogenic soils are unstable and very soluble, as such, which represent highly bioavailable forms[47].

Thus, seabird colonies in polar and subpolar regions could act as real exporting hot spots of P and N to the ocean. More importantly, global change can remobilize these nutrients that have been accumulating through time in soils and sediments, returning them to the sea as a consequence of increasing erosion due to ice thawing and sea level rise, as well as to a presumable increase in pluvial precipitation in Antarctic or sub-Antarctic ecosystems. Similar findings regarding Fe oxyhydroxides have been reported by other authors[51,52].

In summary, ornitheutrophication associated to marine seabird colonies has geochemical and environmental relevance on a global scale. Previous works have demonstrated that seabird colonies can produce important environmental changes at the local level; results obtained in this and the work of Riddick et al.[18], clearly indicate that the magnitudes of the N and P fluxes associated to ornitheutrophication are similar to other processes that are normally considered when calculating global inventories of these two elements (e.g., fishing activities). Hence, N and P budgets can be further improved and refined by including the fluxes of these two elements between the marine environment and the seabird breeding colonies. In addition, it should be noted that a high fraction of the total N and P present in fecal material is readily lixiviated and can become bioavailable.

## Methods

**Global seabird population estimates**. Seabirds are taxonomically a varied group comprising around 3.5% of all birds that depend on the marine environment for at least in part of their life cycle[53,54]. We used a wide range of sources to collate a detailed, spatially explicit database of 320 seabird species, including journal articles, books, and data from international organizations (e.g., BirdLife International and Wetlands International). The data recorded for each colony included bird species, size of the breeding population, and the source from which the information was extracted (Supplementary Data 1). Next, we converted records reported as breeding pairs to total population estimates, assuming that the population includes 30% of non-breeders, a commonly assumed estimate for global seabird studies[55,56]. The seabird database corresponded mainly to the 2002 census. For the sake of clarity, seabird data were arranged in orders: Sphenisciformes (penguins), Procellariiformes (albatross, Shearwaters, and petrels), Pelecaniformes (pelicans, boobies, frigatebirds, tropicbirds, and cormorants), and Charadriiformes (gulls, terns, guillemots, and auks).

**Global N and P excreted by breeding adults**. The global amounts of N excreted by breeding birds and their chicks ($N_{excr(br)}$) were calculated by applying the bioenergetic model used by Riddick et al.[18]. The variables involved in this model include the amount of N excreted by the adult biomass ($M$, in g bird$^{-1}$), nitrogen ($F_{Nc}$, in g N g$^{-1}$ wet mass), and energy ($F_{Ec}$, in kJ g$^{-1}$ wet mass) content of the food, assimilation efficiency of ingested food ($A_{eff}$, in kJ [energy obtained] kJ$^{-1}$ [energy in food]), length of the breeding season ($t_{breeding}$, in days), and the proportion of time spent at the colony during the breeding season ($f_{tc}$, dimensionless parameter):

$$N_{excr(br)} = \frac{9.2M^{0.774}}{F_{Ec}A_{eff}} F_{Nc} t_{breeding} f_{tc} \qquad (1)$$

The data reported by Riddick[57] was used to calculate $t_{breeding}$ and $f_{tc}$, whereas the average values of nitrogen ($F_{Nc} = 0.036$ g N g$^{-1}$) and energy ($F_{Ec} = 6.5$ kJ g$^{-1}$) contents of the seabird diets that were used in Eq. 1 have been reported in different studies[18,21,34]. The term $A_{eff}$ represents the efficiency of conservation of food's energy when it is consumed (kJ obtained by the bird per kJ consumed). An average value of 0.8 is generally assumed for $A_{eff}$[18,21,34,58].

To calculate the amount of P excreted ($P_{excr}$, in g P bird$^{-1}$ y$^{-1}$ in the colony), Eq. 1 was modified by replacing $F_{Nc}$ with the P content of the food ($F_{Pc}$, in g P g$^{-1}$), which according to Furness[58] is equal to 0.0060 g P g$^{-1}$:

$$P_{excr(br)} = \frac{9.2M^{0.774}}{F_{Ec}A_{eff}} F_{Pc} t_{breeding} f_{tc} \qquad (2)$$

**Global N and P excreted by chicks**. The equations used to estimate the annual amounts of N and P excreted by the chicks ($N_{excr(ch)}$ and $P_{excr(ch)}$, respectively) were obtained by using the expressions similar to Eqs. 1 and 2[18]:

$$N_{excr(ch)} = \frac{28.43M_{fledging}^{1.06}}{F_{Ec}A_{eff}} F_{Nc} t_{breeding} f_{tc} \frac{P_{chicks}}{2} \qquad (3)$$

$$P_{excr(ch)} = \frac{28.43M_{fledging}^{1.06}}{F_{Ec}A_{eff}} F_{Pc} t_{breeding} f_{tc} \frac{P_{chicks}}{2} \qquad (4)$$

Chick attendance is estimated as the length of time between hatching and fledging. The $N_{excr(ch)}$ and $P_{excr(ch)}$ values (g N bird$^{-1}$ y$^{-1}$ in the colony or g P bird$^{-1}$ y$^{-1}$ in the colony; Eqs. 3 and 4, respectively) were estimated from the mass of the chick at fledging ($M_{fledging}$, g) and from the breeding productivity ($P_{chicks}$, chicks fledged pair)[18,58,59].

**Global estimates of excreted labile N and P**. Under the general term of labile N and P species were included those readily leachable forms, which in short time spans (e.g., months), can reach marine coastal or continental (lakes, rivers, etc.) waters. Reported average concentrations of nitrogen ($NO_3^-$ + $NH_4^+$) soluble in water or in neutral salts (e.g., KCl, MgCl$_2$ 1 M) that are present in fecal material were considered as labile N species[6,45]. For the case of P, the concentration of phosphate soluble in the extract Mehlich 3 was also included within the labile fraction and considered as bioavailable[6]. All relevant information pertaining labile N and P concentrations were obtained from data reported in the literature (Supplementary Table 3), and from information obtained from analyses of the excrements of *Larus michahellis*[6]. Additionally, volatile P (as phosphine) was assumed to have an average value of $8.8 \times 10^{-6}$ mg kg$^{-1}$ of PH$_3$ in fecal materials[43].

**Map construction**. In order to calculate the amounts of N and P excreted by colonies in different regions, supplementary data provided by Riddick et al. was used, which includes seabird colony locations identified by their geographical coordinates and NH$_3$ emission. Calculation of the amount of N and P excreted by the breeding colonies at worldwide level has limitations due to the scarcity of data regarding their population size, geographic location, and number of species per colony. To solve this problem, the NH$_3$ emission produced by each colony was used, which is proportional to seabird population at each site[18]. First, the mass ratios of excreted N:NH$_3$ and excreted P:NH$_3$ were calculated for each seabird species. Next, these ratios were used to obtain the total N and P excreted by each colony and, finally, all values were added up to obtain the worldwide N and P values. These values were 1.51 times lower than the results obtained, if the worldwide seabird population was used instead. Hence, this correction factor was applied to the N and P values excreted by each colony. Once the amounts of N and P excreted (in kg) in each colony were calculated, map constructions were developed using the *Create Fishnet* tool in ArcGis 10.3 (ESRI) to generate a fishnet grid of square cells with sides of 500 km. The *Spatial Join* tool was used to match more than 3000 colonies to each one of the cells. The *Dissolve* tool was used to group the cells, calculating at the same time the number of points in each cell, and the total amounts of N and P. Finally, the *Field Calculator* tool was used to calculate the density of N and P in each cell by dividing the surface area of each cell by the previous sum.

**Uncertainty in input data**. The variables included in the bioenergetic models are subject to some degree of uncertainty. This is because the behavior of the seabird species depends on the environment where they live, which can undergo changes each year. For example, breeding success shows considerable interannual variation, and the number of days spent in the colony often varies even among populations of the same species[18]. Total population size is also subjected to considerable uncertainty due to differences in census methods and to yearly fluctuations in seabird populations produced by interannual variations in El Niño-Southern Oscillation (in seabird populations in the Pacific and Antarctic regions) or in the North Atlantic Oscillation (in the North Atlantic[56]). It has been suggested that 36% of the error is associated with seabird population estimates, 23% with variations in the composition of the diet, and 13% attributed to non-breeder attendance[18]. However, even considering the highest uncertainties, the main findings of this study will not be affected to a significant extent, since the amounts of N and P that seabirds are capable of mobilizing will still be important in relation to other environmental and anthropogenic processes.

**Data availability**. The authors declare that the data supporting the findings of this study are available within the paper and its supplementary information files.

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

## Acknowledgements

This work was supported by a 2016 BBVA Foundation Grant for Researchers and Cultural Creators, by the Autonomous National Parks Organization (Ref. 041/2010) of the Spanish Ministry for the Environment, Rural and Marine Affairs, and CRETUS strategic group (AGRUP2015/02). S. De La Peña-Lastra benefitted from a predoctoral fellowship from the FPU Programme of the Spanish Ministry of Education and Innovation. The authors thank Esther Sierra for her contribution in the elaboration of Fig. 1. We dedicate this work to the memory of Antonio Sierra and Susiño, two great and irreplaceable friends.

## Author contributions

X.L.O., S.D.L.P.-L., M.A.H.-D., and T.O.F. conceived and developed the research. S.D.L.P.-L. compiled all the information concerning seabird populations and adapted the bioenergetic models. A.P.-A. build the maps showing the distribution of the seabird colonies and the worldwide deposition of nitrogen and phosphorus. All authors contributed equally to the discussion, interpretation of results, and manuscript writing.

## Additional information

**Competing interests:** The authors declare no competing financial interests.

