## [Peer Review File · Nature Communications]

Reviewers' comments:

Reviewer #1 (Remarks to the Author):

SEABIRD COLONIES AS NEW GLOBAL DRIVERS IN THE NITROGEN AND PHOSPHORUS CYCLES

Otero et al.

General Comments

This paper estimates the global seabird population and then uses a bioenergetics model to calculate the total nitrogen and phosphorus excreted by seabirds globally. The global seabird population calculated here of 610 million seabirds is slightly larger than the estimate of 522 million in Riddick et al. (2012), but the data seems to markedly underestimate the penguin population (22 million pairs estimated here compared to 56 million pairs in Riddick (2012)) and this may be because more specific data sources have not been used. The total N excretion, calculated using the method of Wilson et al. (2004) and developed by Riddick et al. (2012), of 591 Gg yr⁻¹ is smaller than the N excretion in Riddick et al. (2012) from Antarctica and Southern Ocean alone (858 Gg N yr⁻¹). The phosphorus excretion rates are the novel part of the paper, however as P excretion is linearly correlated with N excretion, it may be assumed that this is also underestimated.

Even though seabird population data, nitrogen excretion and phosphorus excretion are explicitly presented, this paper very much follows the research undertaken in Riddick et al. (2012) and I am not convinced that the paper is novel enough for publication. The datasets/sources used to calculate the global seabird population are incomplete and do account for the penguin populations in Antarctica and Southern Ocean. As these are the most significant producers of excreta, more effort should be placed in calculating these species' populations. Even though the paper is well written and a great deal of work has gone into producing the seabird database, fundamentally this work, seabird population and N excreta especially, has been done before and more comprehensively than in this study. Potentially the paper would be novel enough for publication if the P transport aspect was developed further with estimates of P flows from the excreta and how this would affect surrounding environments.

Specific comments

L27: remove comment "colonies, via soil"

L29: “Here, an inventory of global seabird populations was carried out and this information was applied to a bioenergetic model to estimate the worldwide amounts of N” This has been done exactly before in Riddick et al. (2012).

L46: Should there be a heading for “Introduction”

L46: “nutrient” instead of “biological”?

L48: What is meant by “large amounts of debris”?

L48: A better example would have been the guano islands off the coast of Peru. The precipitation on the sub-Antarctic islands means that guano is washed away.

L51-60: Interesting but not doesn’t really add to the narrative. Could be condensed into a couple of lines or left out.

L61-63: Which studies? Please add references.

L63: “biological” not necessary.

L66: delete comma “deposition), and”

L67: Add year to Vevers.

L70: Which year? Where about in the Arctic?

L70-75: Verbose, could be contracted.

L77: Specify the cases

L89: Global inputs of N were calculated in Riddick et al. (2012)

L94: A bioenergetics model was used to calculate N and then adapted to calculate P.

L105-114: Database and text reflect an under-estimation of penguin species population. Populations presented in Supplementary Material 1 reflect sources with out-of-date population counts.

L117: Much higher N excretion rates are presented in Riddick et al. (2012)

L123: N excretion rates in Antarctica and Southern Ocean estimated at 858 Gg N yr⁻¹ in Riddick et al. (2012). Discrepancy based on smaller numbers of penguins used in this study. This is carried through the rest of the paper.

L175-188: This should be in the introduction.

L190: Not sure what “a positive geochemical anomaly” means.

L193: Riddick et al. (2012) only estimated the NH₃ emission globally and did not investigate the impact of those emissions. Croft et al. (2016) Contribution of Arctic seabird-colony ammonia to atmospheric particles and cloud-albedo radiative effect, looks more at the effect of NH₃ emissions.

L212: Reformat “(ref. 16)”

L213: Reformat "(ref. 17)"

L215: reformat "(refs. 29, 30)"

L255: As mentioned before, N fluxes have been calculated previously.

L336-338: How were these uncertainties generated?

S Riddick

Reviewer #2 (Remarks to the Author):

Review

Nature Communications 124761_0

SEABIRD COLONIES AS NEW GLOBAL DRIVERS IN THE NITROGEN AND PHOSPHORUS CYCLES

General

The manuscript is very cleanly written – most of my specific comments below are minor typos and grammatical issues (the one annoying one being the lack of scientific names for species in the text). Nonetheless, I appreciated this attention to detail.

My first general thought is that the title is inappropriate – seabird colonies are not “new” drivers, they are only now being recognized as important drivers, but the manuscript would imply that this has been going on for eons, so the title is misleading.

Overall, I liked the concept and theme of the paper and I think it would make a useful, broad contribution to the scientific literature. Topical and interesting.

From an organizational point of view, it was confusing to read along and see the presentation of differences between northern and southern polar species and their relative contributions of N and P,

with the associated argument that despite similar numbers, contributions were substantially different. For someone not familiar with seabirds, this was a paradox. It was only later that the reader could determine that the relative SIZE of the birds in each location differed substantially, which strongly influenced N and P outputs. I think that this entire notion of seabird size needs to be presented earlier in the manuscript, particularly if you want to retain the spatial interpretations of the data. Right now it's difficult to get the key point of that.

However, my main issue with the manuscript is Supplementary Table 1, and particularly the size of seabird populations. Some of the values in there are simply very wrong. For example, the IUCN estimates the global Common Tern *Sterna hirundo* population at 1.6-4,600,00 individuals (https://en.wikipedia.org/wiki/Common_tern), but the authors here use 50,000 as their estimate (3% of the population based on expert opinion and other sources). Arctic terns are estimated at 183,000 birds in the paper, but globally their population is likely much higher, with estimates of ~40,000 birds just in one area of Greenland (https://www.researchgate.net/profile/Carsten_Egevang/publication/232683957_Fluctuating_Breeding_of_Arctic_Terns_Sterna_paradisaea_in_Arctic_and_High-Arctic_Colonies_in_Greenland/links/0c96052f09bb3702e2000000.pdf) and close to 1,000,000 in Eurasia (http://www.jstor.org/stable/30244617?seq=1#page_scan_tab_contents). The ivory gull is listed incredibly at 1,625,000 adults, but that species is red listed by the IUCN at 27,000 individuals (https://en.wikipedia.org/wiki/Ivory_gull; <2% of what the authors used, and these are medium-sized birds)!! Those are just 3 numbers that caught my eye, but many other values for Herring Gull, Western Gull, Black Guillemot, etc. are way, way off. I recognize that numbers are changing all the time and that the authors needed to standardize estimates from somewhere, but a thorough check of those population estimates seems in order, as some of these species estimates alone seem to be off by 90%! I'm less familiar with the penguin numbers and they may be easier to estimate (new techniques with drone photography or even remote sensing), and they likely do carry the lion's share of the N and P in ornithoeutrophication. However, with some of these larger gull population estimates being quite off (and possibly many of the Procellariiformes), I am a bit suspect on the calculations of relative contributions by the different groups of birds.

Thus, I like the idea and approach, but I'm not too sure about the numbers used to input to your models; I know that some of them are really incorrect.

Specific

L27, also in supplementary tables – ornithoeutrophication ... not ornithoeutrophication

L58 – guano, not Guano

L107 – least not lleast

L108 – short-tailed, not shortshort-tailed

L109, L147, L149 – thick-billed, not thickthick-billed

L105-114 – in this section you talk about numbers, but that is a bit deceiving as the Greenland colonies are dominated by little auks which are tiny, and much of the southern regions are dominated by penguins which are large ... might need to clarify here by BIOMASS perhaps?

L127-128 – as per the point above, it is unclear to the uninitiated why the high arctic colonies receive so much less N and P despite similar population sizes – you haven't explained that

L152-153 – You need to provide scientific names for these cormorant species; in fact, check the entire manuscript as this happens in a lot of locations

L160-163 – here you get to the size issue, but I think you need to introduce that earlier (above) as it's unclear

L272 – seabird data were, not was ...

Supplementary Table 1 – many of your population estimates are way off, by almost an order of magnitude, meaning that I cannot be confident in some of your calculations.

Reviewer #3 (Remarks to the Author):

This is a really intriguing contribution, bringing our quantitative understanding of seabird's role in N and P cycling up to a global perspective. This global and comprehensive quantitative view is really what is novel here, and it is timely given that climate and other human impacts on seabirds may impact their distribution/abundance, and thus also impact their role in the N and P cycles. Furthermore, it is important to point out how much of a role they play in these nutrient cycles.

The paper is well-written and documented, and provides a useful context for others working in this area. I have several comments that the authors should address in a revision, none of which are major.

1. The geologic context of this work could be expanded upon. For example, there is not a real evaluation/comparison to seabird N and P fluxes in terms of global fluxes in the main text or figure, and I think that it is important to put this out directly. For example, what is 591 Gg N/yr, or 99 Gg P/yr, in comparison to other fluxes in these elemental cycles? For example, the reactive riverine P flux is around 2,000 Gg P/yr, and thus seabird contributions are about 5% of total riverine flux. This is a big deal, and needs to be made more explicitly clear.

2. The lithology that the seabirds reside in would have a big impact on the net longer-term importance of the fate of excreted N and P (maybe particularly P). This is why the Pacific atolls were such rich phosphate rock resources—excreted P onto a pure carbonate substrate rapidly produces relatively insoluble carbonate fluorapatite. This is a net loss in terms of reactive P...until, that is, we mined it all up for fertilizers! Even a brief reference to the mineralogy of the underlying lithology would help to clarify the longer-term implications of the seabird excretion component of these cycles. Could be inserted in the discussion from line 242 to 248

3. Consider replacing the term “New” in the current title with “Important.” Seabird colonies are not new, but rather they have an important role in global nutrient cycles.

Trivial dits:

Line 32 add “respectively”

Line 89 should be “excrement”

Line 147 typo with Thickthick

Line 182 change to “over tens and hundreds of million years”

Line 474 should be “nutrient”

Line 476 change to “waters it is...”

Line 479 remove “that”

Line 490 change to “simpler cycle”

Line 491 replace “under” with “in”

Line 492 although lixiviated is the proper term, perhaps also add parenthetically “solubilized”

Table 1 has Gg N/yr under the Total P Excreted column heading

Gabriel Filippelli

IUPUI

To Whom it may correspond,

In this document we made our best to address the comments made by the reviewers. Text in **bold italics** represents their comments, which we numbered in progressive order. Text in normal font represents our responses.

We take the opportunity to thank the reviewers for their important contribution to this manuscript.

Reviewer #1 (Remarks to the Author):

General Comments

- 1) ***This paper estimates the global seabird population and then uses a bioenergetics model to calculate the total nitrogen and phosphorus excreted by seabirds globally. The global seabird population calculated here of 610 million seabirds is slightly larger than the estimate of 522 million in Riddick et al. (2012), but the data seems to markedly underestimate the penguin population (22 million pairs estimated here compared to 56 million pairs in Riddick (2012)) and this may be because more specific data sources have not been used.***

Indeed, the most important aspect for the global calculation of N and P deposition is population size. There are several global seabird population estimates reported in the literature, with substantial fluctuations in their values. For example, Brooke (2004) estimated 700 million seabirds and Karpouzi (2007) 900 million individuals. As Riddick (2012) recognizes, the variability in these estimates is a consequence of the sometimes dramatic changes in the size of seabird populations due to climatic events, ecosystem and biological events, and anthropogenic pressures that have been observed at many colonies, indicating the ever-changing nature of these populations (Croxall et al., 2002). Nevertheless, the results are similar between those obtained by us (620 million breeding individuals) and the ones reported by Riddick et al. (2012), who estimated 522 million individuals (261 million pairs, Table A, below). Likewise, the estimates obtained for N deposition are very similar (Table A, below), indicating that our calculations for the deposition of N (and by extension of P) are correct. As we will discuss later, the disagreement among populations of certain species of seabirds (e.g. penguins) was a mix-up produced during the ordering of our database.

Table A. Comparison between our results and those of Riddick (2012).

	Seabird population (Riddick, 2012)	Seabird population In millions of reproductive pairs (this article)	Total N excreted (Riddick, 2012)	Total N excreted from breeding and non-breeding chicks (this paper)	Total P from breeding and non-breeding chicks (this paper)
	Millions of breeding pairs		Gg y ⁻¹		
Antarctica and Southern Ocean	69	81.86	858	845.27	140.88
Greenland and Central America	68	80.68	59	58.12	9.69
World total	261	310	1078	1062	177

The apparent underestimation of seabird populations observed in the manuscript and shown in Table S1 of the manuscript (e.g., discrepancies in penguin populations and species of the genus *Sterna*), with respect to the numbers provided by International data bases or the work of Riddick et al. (2012), was the result of an error inadvertently produced when our database was reordered. Although all population data are correct and updated with respect to previous works, an error was made during the process of ordering our database (Table S1). When the seabird species were reordered in alphabetical order, the Excel program did not properly managed this process so that from row 40 the order of the rows was altered, causing the "Scientific Name" column to be disarranged with respect to the "Breeding Population" column. Table B (below) shows the populations used in our calculations of N and P deposition compared with data from existing international sources. As can be appreciated, they are similar or practically the same.

Table B. Examples of breeding populations of some species or group of species (penguins) used for calculating N and P deposition in the manuscript, compared to population numbers reported in international databases in 2017.

Species	Data base of breeding individuals used in the manuscript (millions of seabirds)	International data base (millions of seabirds)
Total penguin population	44.3	40.9 (3)
Common Tern (Sterna hirundo)	2.06	1.6-4.6 (1)
Arctic Tern (Sterna paradisaea)	2.0	2.0 (2)
Ivory Gull (Pagophila eburnea)	0.015	0.019-0.027 (3)

(1) Wikipedia, according to Reviewer 2

(2) BirdLife International (2016) *Sterna paradisaea*. The IUCN Red List of Threatened Species 2016: e.T22694629A86791057. <http://dx.doi.org/10.2305/IUCN.UK.2016-3.RLTS.T22694629A86791057.en>. Downloaded on 14 July 2017.

(3) BirdLife International (2012). "*Pagophila eburnea*". *IUCN Red List of Threatened Species. Version 2013.2*. International Union for Conservation of Nature. Retrieved 26 November 2013.

2) The total N excretion, calculated using the method of Wilson et al. (2004) and developed by Riddick et al. (2012), of 591 Gg yr⁻¹ is smaller than the N excretion in Riddick et al. (2012) from Antarctica and Southern Ocean alone (858 Gg N yr⁻¹).

The data obtained by us for all seabirds (breeding and non-breeding individuals) was 845 Gg y⁻¹ (Table A), a value slightly lower than the 858 Gg N y⁻¹ obtained by Riddick (2012). However, in our case the seabird population was substantially higher than that of Riddick (2012) (Table A) and, consequently, the excreted N value should also be higher. It is quite possible that the reason for this difference may lie in the size of the penguin population used in the calculations. Based on a new calculation we made of the population for the total of the 18 species of penguins considered in our study and using current sources, we obtained that the total population is 40.9 million individuals (note that they are not pairs) (Table C), a value considerably lower than the 56 million pairs that Reviewer 1 mentions in Question #1 (in our case, we considered individuals, not pairs) and that Riddick used in his calculations. Considering that penguins are the group of species that most contributes to the excretion of N, then we think that this could be the cause of the disagreement between Riddick's calculations (2012) and ours.

Table C. Population of breeding individuals for the 18 species of penguins considered in this study in 2017 (after BirdLife). The differences regarding the size of the populations in our manuscript is due to the fact that the data we used corresponded to censuses of the year 2014 or previous.

Seabird	Species	Population (breeders)	Reference
Emperor Penguin	Aptenodytes forsteri	595000	BirdLife International (2017)
King Penguin	Aptenodytes patagonicus	3200000	BirdLife International (2017)
Rockhopper Penguin	Eudyptes chrysocome	2500000	BirdLife International (2017)
Macaroni Penguin	Eudyptes chrysolophus	12600000	BirdLife International (2017)
Fiordland Penguin	Eudyptes pachyrhynchus	6250	BirdLife International (2017)
Royal Penguin	Eudyptes schlegeli	1700000	BirdLife International (2017)
Erect-crested Penguin	Eudyptes sclateri	150000	BirdLife International (2017)
Snares Penguin	Eudyptes robustus	63000	BirdLife International (2017)
Little Penguin	Eudyptula minor	469760	BirdLife International (2017)
Yellow-eyed Penguin	Megadyptes antipodes	3400	BirdLife International (2017)
Adelie Penguin	Pygoscelis adeliae	7580000	BirdLife International (2017)
Chinstrap Penguin	Pygoscelis antarctica (P. antarcticus)	8000000	BirdLife International (2017)
Gentoo Penguin	Pygoscelis papua	774000	BirdLife International (2017)
African Penguin	Spheniscus demersus	50000	BirdLife International (2017)
Humboldt penguin	Spheniscus humboldti	32000	BirdLife International (2017)
Northern Rockhopper Penguin	Eudyptes moseleyi	480600	BirdLife International (2017)
Magellanic Penguin	Spheniscus magellanicus	2700000	BirdLife International (2017)
Galapagos Penguin	Spheniscus mendiculus	1200	BirdLife International (2017)
TOTAL		40905210	

- 3) ***The phosphorus excretion rates are the novel part of the paper, however as P excretion is linearly correlated with N excretion, it may be assumed that this is also underestimated. Even though seabird population data, nitrogen excretion and phosphorus excretion are explicitly presented, this paper very much follows the research undertaken in Riddick et al. (2012) and I am not convinced that the paper is novel enough for publication. The datasets/sources used to calculate the global seabird population are incomplete and do account for the penguin populations in Antarctica and Southern Ocean. As these are the most significant producers of excreta, more effort should be placed in calculating these species' populations. Even though the paper is well written and a great deal of work has gone into producing the seabird database, fundamentally this work, seabird population and N excreta especially, has been done before and more comprehensively than in this study. Potentially the paper would be novel enough for publication if the P transport aspect was developed further with estimates of P flows from the excreta and how this would affect surrounding environments.***

As we mentioned before, the possible underestimation of seabird populations was the result of an error in our calculations that occurred when the species names were sorted alphabetically in the electronic data sheet. This error was corrected and the corrected values are now integrated in the new manuscript.

Regarding the novelty of our results, we consider that the works of Riddick and collaborators (Riddick, 2012, Riddick et al., 2012, 2017) were extremely valuable in relation to the importance of seabirds in the global cycle of N and in the emissions of NH₃ into the atmosphere. However, our work presents other novel aspects not contemplated by the series of works of Riddick and collaborators (Riddick, 2012, Riddick et al., 2012, 2017). The work we present in the manuscript, in addition to the overall contribution of N that Riddick et al (2012) made, also provides relevant information on the contribution of each of the 320 species considered. In addition, our work is the first scientific document to show the worldwide distribution of seabird excretions in detailed meshes (500x500 km; Fig 2 of the manuscript). Finally, our results for the excretion of P are the first to be performed both globally and by colonies.

In our new manuscript version, a new contribution has been made, which consists of estimating the concentrations of N and P that can be easily leached (solubilized) and transported from seabird colonies to coastal waters or to inland surface waters (lagoons, rivers, etc.) in the short term (months). For these calculations, data obtained from our work on *Larus michahellis* (before *L. cachinnans*) colonies in the Atlantic Islands National Park (Otero & Fernández Sanjurjo, 2000; Otero et al., 2015) were used, as well as literature data on the concentrations of labile N and P present in seabird excrement. We considered that the labile forms of these two elements were represented by ammonium (NH₄⁺) and nitrate (NO₃⁻), both present in seabird excrements. The NO₃⁻ species are barely adsorbed by the soil colloidal system, a mechanism that rarely occurs with the cationic species NH₄⁺; however, ammonia is rapidly oxidized to nitrate by nitrifying bacteria (Otero & Fernández Sanjurjo, 2000, De Peña Lastra 2012). As for the leaching of P, it is assumed that the adsorption capacity of P by the seabird colonies soil is negligible or very low because the colonies are generally located directly on rocks or on shallow, stony soils and, frequently, sandy with a colloidal system saturated with P (e.g., Otero et al., 2015). For phosphorus we also calculated the phosphorus emitted into the atmosphere as phosphine (PH₃) from the scarce data available (e.g., Zhu et al., 2006). In general, the concentrations of the labile forms of N and P ranged from 12-21% of the total concentration present in the fresh faecal material.

In addition, a global water balance was calculated (data not shown in the manuscript) using precipitation and temperature data from meteorological stations located close to the seabird colonies to determine the areas of the colonies where excess rain necessary to solubilize the labile N and P is present. Our results showed that in most of the colonies there is an excess of precipitation, at least in some month along the year, that allows the leaching of these two elements. Only in the extremely arid coasts, known as guanera areas (e.g., coasts of Peru, California, Namibia), this process may be irrelevant.

Our data show that the concentration of labile N that can be mobilized towards the coasts, rivers or lakes was 71.8 Gg y^{-1} , although this result does not include the NH_3 emitted into the atmosphere that was already calculated by Riddick et al. (2012). For P, its leachable concentration was 18 Gg y^{-1} , while the P emitted to the atmosphere was negligible, both at global and local scales (1.5 kg y^{-1}).

Table D. Total and labile annual masses of excreted P y N.

Order	Population of chicks and breeders (millions of seabirds)	Total P (Gg y^{-1})	Labile P (Gg y^{-1})	Total N (Gg y^{-1})	Labile N* (Gg y^{-1})
Charadriiformes	291	19	5.07	116	9.79
Pelecaniformes	31	9	1.45	51	5.82 (1)
Procellariiformes	424	20	3.22 (1)	117	21.20
Sphenisciformes	59	51	8.22 (1)	307	35.03
TOTAL	804	99	17.96	591	71.84

*Represents $\text{NH}_4^+ + \text{NO}_3^-$

(1) Reported values not available. Calculations were made using the average values of labile N and P forms from all available data on seabirds.

Specific comments

4) **L27: remove comment “colonies, via soil”**

We decided to keep this term in the new manuscript because it gives a wider and detailed idea of the eutrophication effect produced by seabird colonies.

5) **L29: “Here, an inventory of global seabird populations was carried out and this information was applied to a bioenergetic model to estimate the worldwide amounts of N” This has been done exactly before in Riddick et al. (2012).**

To clarify this part, the following text was included in the Section of Materials and methods of the new manuscript [lines 304-305]:

“The global amounts of N excreted by breeding birds and their chicks ($N_{\text{excr(br)}}$) were calculated by applying the bioenergetic model used by Riddick et al.¹⁸”

6) **L46: Should there be a heading for “Introduction”**

Nature Communications explicitly states that submitted manuscripts should “Avoid ‘Introduction’ as a heading.”

7) **L46: “nutrient” instead of “biological”?**

Similarly to other authors (e.g., Michelutti, 2009), we decided to keep “biological”, since we think it is a more precise term than “nutrient” for defining the effect produced by seabirds.

Michelutti, N. Seabird-driven shifts in Arctic pond ecosystems. *Proc. R. Soc. B* **276**, 591–596 doi:10.1098/rspb.2008.1103 (2009).

8) **L48: What is meant by “large amounts of debris”?**

The meaning of “large amounts of debris” was explained in lines 46-49 of the old manuscript [lines 47-50 of the new manuscript]:

"For example, faecal material in penguin colonies from Marion Island represented about 85% of all organic debris deposited on the substrate, with huge amounts (~100 Mg dry weight) accumulating in the colonies during the nesting season"

9) L48: A better example would have been the guano islands off the coast of Peru. The precipitation on the sub-Antarctic islands means that guano is washed away.

The accumulation of guano over the years in arid coastal areas is the best example, but in this case reference is made to the effect of seabirds during the rearing period, which is important at all latitudes, although an important portion of the debris material is later removed by runoff or leaching, as we show in Figure 1 of the new manuscript.

10) L51-60: Interesting but not doesn't really add to the narrative. Could be condensed into a couple of lines or left out.

We agree with the Reviewer. Several sentences from the paragraph were eliminated, condensing it in the process. The relevant text was changed from [lines 51-60 of the old manuscript]:

"This faecal material, known as *guano* (a term derived from the Quechuan word for dung or animal excrement), contains high concentrations of macro and micronutrients^{4,6}. This fact explains why guano has been used since ancient times as a natural fertilizer to enhance agricultural productivity^{7,8}. Guano-producing birds were recognized as protected species by the Inca Empire (Peru), and anyone found disturbing breeding colonies could be sentenced to death⁷. However, it was until the second half of the 19th century that researchers in Napoleonic Europe identified Peruvian guano and nitrates from Chile as the richest sources of N ever discovered⁸. Thus, Guano from seabird colonies fueled the great demand for food (crops and meat) as well as the production of explosives that happened in the northern hemisphere, thus promoting the development and growth of the European population⁷."

To [lines 50-55 of new manuscript]:

"This faecal material, known as *guano* (a term derived from the Quechuan word for dung or animal excrement), contains high concentrations of macro and micronutrients^{4,6}, and has been used since ancient times as a natural fertilizer^{7,8}. However, it was until the second half of the 19th century that European researchers identified Peruvian guano as one of the richest sources of N ever discovered⁸. Thus, guano from seabird colonies fueled the great demand for food that happened in the northern hemisphere, promoting the development and growth of the European population⁷."

11) L61-63: Which studies? Please add references.

As suggested by the Reviewer, several relevant references were added to the sentence [lines 56-58 of the new manuscript]:

"In addition to the eco-historical importance of guano, many researchers have investigated the effect that seabird colonies have on the biogeochemical processes and vegetation ecology at different geographical scales (local and regional; Fig.1)^{4,6,9-13}."

4. Sobey, D. G., Kenworthy, J. B. The relationship between herring gulls and the vegetation of their breeding colonies. *Journal of Ecology* **67**, 469-496 (1979).
6. Otero, X.L. *et al.* Phosphorus in seagull colonies and the effect on the habitats. The case of yellow-legged gulls (*Larus michahellis*) in the Atlantic Islands National Park (Galicia-NW Spain). *Science of the Total Environment* **532**, 383-397 (2015).
9. Vidal, E., Médail, F., Tatoni, T., Roche, P. & Bonnet, V. Seabirds drive plant species turnover on small Mediterranean islands at the expense of native taxa. *Oecologia* **122**, 427-434 (2000).
10. Sanchez-Piñero, F. & Polis, G. A. Bottom-up dynamics of allochthonous input: direct and indirect effects of seabirds on islands. *Ecology* **81**, 3117-3132 (2000).
11. Vevers, H. G. The land vegetation of Ailsa Craig. *Journal of Ecology* **24**, 424-445 (1936).

12. Russell, R. S., Cutler, D. W., Jacobs, S.E., King, A. & Pollard, A.G. Physiological and ecological studies on an Arctic vegetation: II, The development of vegetation in relation to nitrogen supply and soil microorganisms on Jan Mayen Island. *Journal of Ecology* **28**, 429-454 (1940).
13. Lorrain et al., Seabird supply nitrogen to reef-building corals on remote Pacific islets. *Scientific Reports* **7**, 3721, doi:10.1038/s41598-017-03781-y (2017).

12) L63: “biological” not necessary.

The term “biological” was eliminated from the sentence, as suggested by the Reviewer.

13) L66: delete comma “deposition), and”

The text was corrected as suggested by the Reviewer.

14) L67: Add year to Vevers.

The year (1936) was added to Vevers, as suggested by the Reviewer [line 62 of the new manuscript].

15) L70: Which year? Where about in the Arctic?

The text was changed from [lines 70-72 of the old manuscript]:

“A few years later, in a study carried out in the Arctic region, Russell et al.¹² reported that vegetation was generally sparse and the amount of nitrogen derived from dead tissues was accordingly small.”

To [lines 65-68 of the new manuscript]:

“A few years later, in a study carried out in Jan Mayen Island in the Arctic region, Russell *et al.*¹² reported in 1940 that the amount of nitrogen derived from dead plant tissues was small due to the low temperatures in the region, which restricted the activity of the microorganisms responsible for decomposing organic matter.”

16) L70-75: Verbose, could be contracted.

We agree with the Reviewer. The paragraph was changed from [lines 70-75 of the old manuscript]:

“A few years later, in a study carried out in the Arctic region, Russell et al.¹² reported that vegetation was generally sparse and the amount of nitrogen derived from dead tissues was accordingly small. These characteristics were a consequence of the low temperatures present in the region, which could restrict the activity of the microorganisms responsible for decomposing organic matter; however, the nitrogen supplied by seabird colonies would exert an important effect on the development of Arctic plant communities.”

To [lines 65-69 of the new manuscript]:

“A few years later, in a study carried out in Jan Mayen Island in the Arctic region, Russell *et al.*¹² reported in 1940 that the amount of nitrogen derived from dead plant tissues was small due to the low temperatures in the region, which restricted the activity of the microorganisms responsible for decomposing organic matter. However, the nitrogen supplied by seabird colonies would exert an important effect on the development of Arctic plant communities and the appearance of new plant taxa.”

17) L77: Specify the cases

Several locations where seabird colonies have significantly transformed the environment have now been included in the new manuscript. The relevant text was changed from [lines 77-80 of the old manuscript]:

“Although in some cases seabird colonies have profoundly altered the biogeochemical processes that occur in coastal surface systems (soils, sediments, waters) and have transformed plant communities, most studies revealing biogeochemical and ecological alterations have been mostly of local interest^{4,9,10} (Fig. 1).”

To [lines 70-74 of the new manuscript]

“Although in some cases seabird colonies have profoundly altered the biogeochemical processes that occur in coastal surface systems (soils, sediments, waters) and have transformed plant communities (for example, Mediterranean and Atlantic islands, North-East Scottish coast, Pacific reef corals), most studies revealing biogeochemical and ecological alterations have been mostly of local interest^{6,9-14} (Fig. 1).”

18) L89: Global inputs of N were calculated in Riddick et al. (2012)

Indeed, Riddick et al. (2012) were the first to perform the calculation of the total deposition of N, but not the one of P. Hence, the allusion to N was eliminated, but the one to phosphorus was left. Hence, the text from the old manuscript was changed from [lines 89-90]:

“However, global inputs of N and P from seabird colonies have not yet been estimated.”

To [lines 83-84 of the new manuscript]:

“However, global inputs of P from seabird colonies have not yet been estimated.”

19) L94: A bioenergetics model was used to calculate N and then adapted to calculate P.

To address the Reviewer’s concern, the text was changed from [Lines 93-95]:

“A bioenergetic model (proposed by Wilson et al.¹⁸, and later used by Riddick et al.¹⁶) was adapted to calculate the amounts of N and P excreted by reproducing seabirds and their chicks in seabird colonies worldwide.”

To [lines 87-89 of the new manuscript]:

“A bioenergetics model (proposed by Wilson *et al.*²¹, and later used by Riddick *et al.*¹⁸) was used to calculate the amounts of N, and then adapted to calculate the quantities of P excreted by reproducing seabirds and their chicks in colonies worldwide.”

20) L105-114: Database and text reflect an under-estimation of penguin species population. Populations presented in Supplementary Material 1 reflect sources with out-of-date population counts.

This observation was answered in detail in our response to Observation (1).

21) L117: Much higher N excretion rates are presented in Riddick et al. (2012)

This observation was answered in detail in our response to Observation (2).

22) L123: N excretion rates in Antarctica and Southern Ocean estimated at 858 Gg N y⁻¹ in Riddick et al. (2012). Discrepancy based on smaller numbers of penguins used in this study. This is carried through the rest of the paper.

This observation was answered in detail in our response to Observation (2).

23) L175-188: This should be in the introduction.

We tend to agree with the Reviewer, but we also consider that it is more convenient to start the Discussion section by referring to general aspects of the N and P cycles, whereas in the Introduction we were more focused on the effects of seabirds on the natural environment.

24) L190: Not sure what “a positive geochemical anomaly” means.

To address the Reviewer’s concern, the text was changed from [lines 189-191 of the old manuscript]:

“Taking the above into account, various authors have suggested that seabird colonies represent a positive geochemical anomaly regarding the concentrations of N and P present in soils, sediments and water^{1,6,28}, hereafter termed ornithoetrophication (Fig. 1).”

To [lines 197-199 of the new manuscript]:

“Taking the above into account, various authors have suggested that seabird colonies represent a positive geochemical anomaly (i.e., above background values) regarding the concentrations of N and P present in soils, sediments and water^{1,6,13,33}, hereafter termed *ornithoetrophication* (Fig. 1).”

25) L193: Riddick et al. (2012) only estimated the NH₃ emission globally and did not investigate the impact of those emissions. Croft et al. (2016) Contribution of Arctic seabird-colony ammonia to atmospheric particles and cloud-albedo radiative effect, looks more at the effect of NH₃ emissions.

We agree with the Reviewer. We mention that the works of Riddick and collaborators, and especially Riddick et al. (2012), represented fundamental scientific contributions to understand the importance of seabird colonies in the global N cycle. We also introduced the work of Croft et al. (2016) as a clear example of the large-scale influence of seabird colonies on the environment. Hence, the text was changed from [lines 193-196 of the old manuscript]:

“Riddick et al.¹⁶ extrapolated the impact of atmospheric emissions of NH₃ from the mineralization of uric acid present in seabird excrements. These authors reported that global emissions from seabird excrements may range between 97 and 442 Gg NH₃ y⁻¹, making them an environmentally relevant process²⁷.”

To [lines 201-204 of the new manuscript]:

“Riddick *et al.*¹⁸ extrapolated the impact of atmospheric emissions of NH₃ from the mineralization of uric acid present in seabird excrements. These and other authors reported that global emissions from seabird excrements may range between 97 and 442 Gg NH₃ y⁻¹, making them an environmentally relevant process³³⁻³⁵.”

18. Riddick S.N., *et al.* The global distribution of ammonia emissions from seabird colonies. *Atmospheric Environment* **55**, 319-327 (2012)

33. Sutton M.A. *et al.* Towards a climate-dependent paradigm of ammonia emission and deposition, *Phil. Trans. R. Soc. B* 368, 20130166, <http://dx.doi.org/10.1098/rstb.2013.0166> (2013).

34. Blackall, T. D. *et al.* Ammonia emissions from seabird colonies, *Geophysical Research Letters* **34**, 5-17 (2007).

35. Croft, B., Wentworth, G.R., Martin, R.V., Leitch, W.R., Murphy, J.G., Murphy, B.N., Kodros, J.K., Abbatt, J.P.D., Pierce, J.R. Contribution of Arctic seabird-colony ammonia to atmospheric particles and cloud-albedo radiative effect. *Nat. Commun.* **7**, 13444 doi: 10.1038/ncomms13444 (2016).

26) L212: Reformat “(ref. 16)”

To address the Reviewer’s concern, the text was changed from [lines 211-215 of the old manuscript]:

“Thus, for a Macaroni Penguin colony these inputs can be as high as 114,240 kg N ha⁻¹ y⁻¹ (ref. 16), whereas those for a Northern Gannet colony can reach 52,200 kg N ha⁻¹ y⁻¹ (ref. 27). These values are the highest known for the Earth’s surface, representing between 500 and >1100 times the total annual inputs of N from agriculture (~65-80 kg N ha⁻¹; refs. 29,30).”

To [lines 219-223 of the new manuscript]:

“Thus, for a Macaroni Penguin colony these inputs can be as high as¹⁸ 114,240 kg N ha⁻¹ y⁻¹, whereas those for a Northern Gannet colony can reach³³ 52,200 kg N ha⁻¹ y⁻¹. These values are the highest known for the Earth’s surface, representing between 500 and >1100 times the total annual inputs of N from agriculture^{36,37} (~65-80 kg N ha⁻¹).”

27) L213: Reformat “(ref. 17)”

To address the Reviewer’s concern, the text was changed from [lines 211-215 of the old manuscript]:

“Thus, for a Macaroni Penguin colony these inputs can be as high as 114,240 kg N ha⁻¹ y⁻¹ (ref. 16), whereas those for a Northern Gannet colony can reach 52,200 kg N ha⁻¹ y⁻¹ (ref. 27). These values are the highest known for the Earth’s surface, representing between 500 and >1100 times the total annual inputs of N from agriculture (~65-80 kg N ha⁻¹; refs. 29,30).”

To [lines 219-223 of the new manuscript]:

“Thus, for a Macaroni Penguin colony these inputs can be as high as¹⁸ 114,240 kg N ha⁻¹ y⁻¹, whereas those for a Northern Gannet colony can reach³³ 52,200 kg N ha⁻¹ y⁻¹. These values are the highest known for the Earth’s surface, representing between 500 and >1100 times the total annual inputs of N from agriculture^{36,37} (~65-80 kg N ha⁻¹).”

28) L215: reformat “(refs. 29, 30)”

To address the Reviewer’s concern, the text was changed from [lines 211-215 of the old manuscript]:

“Thus, for a Macaroni Penguin colony these inputs can be as high as 114,240 kg N ha⁻¹ y⁻¹ (ref. 16), whereas those for a Northern Gannet colony can reach 52,200 kg N ha⁻¹ y⁻¹ (ref. 27). These values are the highest known for the Earth’s surface, representing between 500 and >1100 times the total annual inputs of N from agriculture (~65-80 kg N ha⁻¹; refs. 29,30).”

To [lines 219-223 of the new manuscript]:

“Thus, for a Macaroni Penguin colony these inputs can be as high as¹⁸ 114,240 kg N ha⁻¹ y⁻¹, whereas those for a Northern Gannet colony can reach³³ 52,200 kg N ha⁻¹ y⁻¹. These values are the highest known for the Earth’s surface, representing between 500 and >1100 times the total annual inputs of N from agriculture^{36,37} (~65-80 kg N ha⁻¹).”

29) L255: As mentioned before, N fluxes have been calculated previously.

We agree with the Reviewer and now we included the reference of Riddick et al. (2012) in the new manuscript [lines 278-280]:

“Previous works have demonstrated that seabird colonies can produce important environmental changes at the local level; results obtained in this and the work of Riddick *et al.*¹⁸, clearly indicate that the magnitudes of the N and P fluxes ...”

30) L336-338: How were these uncertainties generated?

As indicated in the old manuscript [lines 368-381], the uncertainties were obtained from Riddick *et al.* (2012) and references therein.

Reviewer #2 (Remarks to the Author):

Review

Nature Communications 124761_0

SEABIRD COLONIES AS NEW GLOBAL DRIVERS IN THE NITROGEN AND PHOSPHORUS CYCLES

General

31) The manuscript is very cleanly written – most of my specific comments below are minor typos and grammatical issues (the one annoying one being the lack of scientific names for species in the text). Nonetheless, I appreciated this attention to detail.

The scientific names of all mentioned seabird species are now included in the new manuscript; however, please note that all of them were already listed in our old Table S1. We did not include the scientific names in the old manuscript because we followed the example of other articles, where they did not include them either.

32) My first general thought is that the title is inappropriate – seabird colonies are not “new” drivers, they are only now being recognized as important drivers, but the manuscript would imply that this has been going on for eons, so the title is misleading. Overall, I liked the concept and theme of the paper and I think it would make a useful, broad contribution to the scientific literature. Topical and interesting.

We agree with the Reviewer. The title was changed from:

“SEABIRD COLONIES AS NEW GLOBAL DRIVERS IN THE NITROGEN AND PHOSPHORUS CYCLES”

To:

“SEABIRD COLONIES AS IMPORTANT GLOBAL DRIVERS IN THE NITROGEN AND PHOSPHORUS CYCLES”

33) From an organizational point of view, it was confusing to read along and see the presentation of differences between northern and southern polar species and their relative contributions of N and P, with the associated argument that despite similar numbers, contributions were substantially different. For someone not familiar with seabirds, this was a paradox. It was only later that the reader could determine that the relative SIZE of the birds in each location differed substantially, which strongly influenced N and P outputs. I think that this entire notion of seabird size needs to be presented earlier in the manuscript, particularly if you want to retain the spatial interpretations of the data. Right now it’s difficult to get the key point of that.

We agree with the Reviewer. To address this observation, the following paragraph was added to the new manuscript [lines 105-118]:

“Global distribution of the seabird colonies showed that they were distributed mainly in the polar zones (Fig. 2), with more than half of the total population concentrated in Antarctica and its sub-Antarctic islands (213 million) and in Greenland and Svalbard islands (209 million). However, despite a similar distribution in the total number of seabirds between polar zones, it should be taken into consideration that large population sizes do not necessarily correspond to large nutrient excretions¹⁸. The differences between species’ body masses and length of the breeding seasons are the main reasons why nutrient excretions in Antarctica and its sub-Antarctic islands were far superior than the ones obtained for Greenland and Svalbard islands. For example, species from the Arctic zone are small in size and weight, with the body masses of the two most abundant species (Little Auk and Least Auklet) in the order of 0.15-0.18²³ and ~0.08 kg²⁴, respectively. However, an important portion of the species present in Antarctica and its sub-Antarctic islands are big in size and weight, as is the case with the Chinstrap (*Pygoscelis antarcticus*, 3-5 kg²⁵) and Emperor (*Aptenodytes forsteri*: 22-37 Kg²⁶) penguins. These differences in body mass have a dramatic effect on the quantity of excreted N and P, as discussed in the following section.”

34) However, my main issue with the manuscript is Supplementary Table 1, and particularly the size of seabird populations. Some of the values in there are simply very wrong. For example, the IUCN estimates the global Common Tern *Sterna hirundo* population at 1.6-4,600,00 individuals (https://en.wikipedia.org/wiki/Common_tern), but the authors here use 50,000 as their estimate (3% of the population based on expert opinion and other sources). Arctic terns are estimated at 183,000 birds in the paper, but globally their population is likely much higher, with estimates of ~40,000 birds just in one area of Greenland (https://www.researchgate.net/profile/Carsten_Egevang/publication/232683957_Fluctuating_Breeding_of_Arctic_Terns_Sterna_paradisaea_in_Arctic_and_High-Arctic_Colonies_in_Greenland/links/0c96052f09bb3702e2000000.pdf) and close to 1,000,000 in Eurasia (http://www.jstor.org/stable/30244617?seq=1#page_scan_tab_contents). The ivory gull is listed incredibly at 1,625,000 adults, but that species is red listed by the IUCN at 27,000 individuals (https://en.wikipedia.org/wiki/Ivory_gull; <2% of what the authors used, and these are medium-sized birds!! Those are just 3 numbers that caught my eye, but many other values for Herring Gull, Western Gull, Black Guillemot, etc. are way, way off. I recognize that numbers are changing all the time and that the authors needed to standardize estimates from somewhere, but a thorough check of those population estimates seems in order, as some of these species estimates alone seem to be off by 90%! I'm less familiar with the penguin numbers and they may be easier to estimate (new techniques with drone photography or even remote sensing), and they likely do carry the lion's share of the N and P in ornithoeutrophication. However, with some of these larger gull population estimates being quite off (and possibly many of the Procellariiformes), I am a bit suspect on the calculations of relative contributions by the different groups of birds. Thus, I like the idea and approach, but I'm not too sure about the numbers used to input to your models; I know that some of them are really incorrect.

Our calculations are based on updated seabird populations obtained from different sources of recognized prestige and international solvency. The apparent underestimation of seabird populations observed in the manuscript and shown in Table S1 of the manuscript (e.g., discrepancies in penguin populations and species of the genus *Sterna*), with respect to the numbers provided by International data bases or the work of Riddick et al. (2012), was the result of an error inadvertently produced when our database was reordered. Although all population data are correct and updated with respect to previous works, an error was made during the process of ordering our database (Table S1). When the seabird species were reordered in alphabetical order, the Excel program did not properly managed this process so that from row 40 the order of the rows was altered, causing the "Scientific Name" column to be disarranged with respect to the "Breeding Population" column. Table B (below) shows the populations used in our calculations of N and P deposition compared with data from existing international sources. As can be appreciated, they are similar or practically the same.

Table B. Examples of breeding populations of some species or group of species (penguins) used for

calculating N and P deposition in the manuscript, compared to population numbers reported in international databases in 2017.

Species	Data base of breeding individuals used in the manuscript (millions of seabirds)	International data base (millions of seabirds)
Total penguin population	44.3	40.9 (3)
Common Tern (Sterna hirundo)	2.06	1.6-4.6 (1)
Artic Tern (Sterna paradisaea)	2.0	2.0 (2)
Ivory gull (Pagophila eburnea)	0.15	0.19-0.27 (3)

(1) Wikipedia, according to Reviewer 2

(2) BirdLife International (2016) *Sterna paradisaea*. The IUCN Red List of Threatened Species 2016: e.T22694629A86791057. <http://dx.doi.org/10.2305/IUCN.UK.2016-3.RLTS.T22694629A86791057.en>. Downloaded on 14 July 2017.

(3) BirdLife International (2012). "*Pagophila eburnea*". *IUCN Red List of Threatened Species. Version 2013.2*. International Union for Conservation of Nature. Retrieved 26 November 2013.

Specific

35) L27, also in supplementary tables – ornithotrophication ... not ornitoeutrophication

The term “ornithoeutrophication” was changed to “ornithotrophication” throughout all the new manuscript.

36) L58 – guano, not Guano

The text was corrected as suggested by the Reviewer.

37) L107 – least not lleast

The text was corrected as suggested by the Reviewer.

38) L108 – short-tailed, not shortshort-tailed

The text was corrected as suggested by the Reviewer.

39) L109, L147, L149 – thick-billed, not thickthick-billed

The text was corrected as suggested by the Reviewer.

40) L105-114 – in this section you talk about numbers, but that is a bit deceiving as the Greenland colonies are dominated by little auks which are tiny, and much of the southern regions are dominated by penguins which are large ... might need to clarify here by BIOMASS perhaps?

We agree with the Reviewer. To address this observation, the following paragraph was added to the new manuscript [lines 105-118]:

“Global distribution of the seabird colonies showed that they were distributed mainly in the polar zones (Fig. 2), with more than half of the total population concentrated in Antarctica and its sub-Antarctic islands (213 million) and in Greenland and Svalbard islands (209 million). However, despite a similar distribution in the total number of seabirds between polar zones, it should be taken into consideration that large population sizes do not necessarily correspond to large nutrient excretions¹⁸. The differences between species’ body masses and length of the breeding seasons are the main reasons why nutrient excretions in Antarctica and its sub-Antarctic islands were far superior than the ones obtained for Greenland and Svalbard islands. For example, species from the Arctic zone are small in size and weight, with the body masses of the two most abundant species (Little Auk and Least Auklet) in the order of 0.15-0.18²³ and ~0.08 kg²⁴, respectively. However, an important portion of the species present in Antarctica and its sub-Antarctic islands are big in size and weight, as is the case with the Chinstrap (*Pygoscelis antarcticus*, 3-5 kg²⁵) and Emperor (*Aptenodytes forsteri*: 22-37

Kg²⁶) penguins. These differences in body mass have a dramatic effect on the quantity of excreted N and P, as discussed in the following section.”

41) L127-128 – as per the point above, it is unclear to the uninitiated why the high arctic colonies receive so much less N and P despite similar population sizes – you haven’t explained that

See response to the preceding Comment #40.

42) L152-153 – You need to provide scientific names for these cormorant species; in fact, check the entire manuscript as this happens in a lot of locations

The scientific names of all mentioned seabird species are now included in the new manuscript; however, please note that all of them were already listed in our old Table S1. We did not include the scientific names in the old manuscript because we followed the example of other articles, where they did not include them either.

43) L160-163 – here you get to the size issue, but I think you need to introduce that earlier (above) as it’s unclear

To explain this apparent contradiction (see also Comment #33), the following paragraph was added to the new manuscript [lines 105-118 of the new manuscript]:

“Global distribution of the seabird colonies showed that they were distributed mainly in the polar zones (Fig. 2), with more than half of the total population concentrated in Antarctica and its sub-Antarctic islands (213 million) and in Greenland and Svalbard islands (209 million). However, despite a similar distribution in the total number of seabirds between polar zones, it should be taken into consideration that large population sizes do not necessarily correspond to large nutrient excretions¹⁸. The differences between species’ body masses and length of the breeding seasons are the main reasons why nutrient excretions in Antarctica and its sub-Antarctic islands were far superior than the ones obtained for Greenland and Svalbard islands. For example, species from the Arctic zone are small in size and weight, with the body masses of the two most abundant species (Little Auk and Least Auklet) in the order of 0.15-0.18²³ and ~0.08 kg²⁴, respectively. However, an important portion of the species present in Antarctica and its sub-Antarctic islands are big in size and weight, as is the case with the Chinstrap (*Pygoscelis antarcticus*, 3-5 kg²⁵) and Emperor (*Aptenodytes forsteri*: 22-37 Kg²⁶) penguins. These differences in body mass have a dramatic effect on the quantity of excreted N and P, as discussed in the following section”

44) L272 – seabird data were, not was ...

The text was corrected as suggested by the Reviewer.

45) Supplementary Table 1 – many of your population estimates are way off, by almost an order of magnitude, meaning that I cannot be confident in some of your calculations.

Table S1 was corrected for the unintentional error produced when the data in the Excel sheet was rearranged (see complete explanation in the response to Comment #34.). Our new Table S1 now shows each species correctly matching their respective populations. We deeply regret this error.

Reviewer #3 (Remarks to the Author):

This is a really intriguing contribution, bringing our quantitative understanding of seabird’s role in N and P cycling up to a global perspective. This global and comprehensive quantitative view is really what is novel here, and it is timely given that climate and other human impacts on seabirds

may impact their distribution/abundance, and thus also impact their role in the N and P cycles. Furthermore, it is important to point out how much of a role they play in these nutrient cycles.

The paper is well-written and documented, and provides a useful context for others working in this area. I have several comments that the authors should address in a revision, none of which are major.

46) The geologic context of this work could be expanded upon. For example, there is not a real evaluation/comparison to seabird N and P fluxes in terms of global fluxes in the main text or figure, and I think that it is important to put this out directly. For example, what is 591 Gg N/yr, or 99 Gg P/yr, in comparison to other fluxes in these elemental cycles? For example, the reactive riverine P flux is around 2,000 Gg P/yr, and thus seabird contributions are about 5% of total riverine flux. This is a big deal, and needs to be made more explicitly clear.

We fully agree with the Reviewer and for this reason we now have included in the supplementary material the new Tables S2 and S3, which show the quantities of N and P present in the main geochemical compartments, as well as the flows between them. As indicated in the new manuscript, the N and P flows from the oceans to the continents caused by the activity of seabirds are similar in size to those usually considered within the cycle of both elements. We think that these results represent one of the main contributions of our work.

Following the recommendations of the Reviewer, we modified the discussion to give greater emphasis to the comparison between the fluxes of N and P usually considered within the geochemical cycle of these two elements with those obtained in this work. For example, lines 204-207 of the old manuscript were changed from:

“Similar results were obtained for P, with its flow from marine to terrestrial environments attributed to seabirds (breeding seabirds: 0.10×10^3 Gg P y^{-1} ; total population: 0.63×10^3 Gg P y^{-1}) being of similar magnitude than those occurring between oceanic waters and atmosphere, or those due to fishing activities (0.31×10^3 and 0.32×10^3 Gg P y^{-1} , respectively; Supplementary Table 3).”

To [lines 211-215 of the new manuscript]:

“Similar results were obtained for P, with its flow from marine to terrestrial environments attributed to seabirds (breeding seabirds: 0.10×10^3 Gg P y^{-1} ; total population: 0.63×10^3 Gg P y^{-1}) being of similar magnitude than those occurring between oceanic waters and atmosphere (0.31×10^3 Gg P y^{-1}), those produced by fishing activities (0.32×10^3 Gg P y^{-1}) or those attributed to the dissolved inorganic P flux of rivers (0.8 - 1.4×10^3 Gg P y^{-1}) (Supplementary Table 3).”

And the following sentence was changed from [lines 234-238 of the old manuscript]:

“However, the main differences relative to N are that P lacks a stable gaseous phase³⁶ and that mobilization of P via leaching is severely restricted due to its adsorption to soil colloids⁶ (Fig. 1). Thus, although annual losses of N may occur via leaching^{37,38}, P accumulates in the soil, with extremely high values maintained over time⁶.”

To [lines 253-261 of the new manuscript]:

“However, the majority of the seabird colonies' soils are located on rocky substrates, with shallow and sandy soils or where the occurrence of permafrost near the soil surface has a strong regulatory effect on leaching and development processes⁴⁷⁻⁴⁸. Under these conditions, a rapid P soil saturation is eventually reached and this element is then lixiviated towards coastal or continental waters⁶. In this sense, the results obtained in this work show that 21% (0.02×10^3 Gg y^{-1}) of the total excreted P is readily lixiviated. Hence, the fraction of P excreted by seabirds can be considered as relevant relative to the total P flux to the oceans since it represents 2.5% and 10% of the total dissolved fluxes of inorganic and organic P by rivers, respectively (Supplementary Table 3).”

In addition, more updated data have been included for some of the flows shown in the new Table S3 of the supplementary material, where we incorporated information reported by Paytan and McLaughlin (2007) and Compton et al. (2000).

Finally, the following paragraph was added to the new manuscript [lines 177-180]:

“On the other hand, the amounts of excreted labile forms of N and P (those that can be readily dissolved) were 72.5 Gg y^{-1} and 21.8 Gg y^{-1} , respectively, with the highest values corresponding to the Sphenisciformes (35.0 Gg N y^{-1} and 10.9 Gg P y^{-1} , respectively), followed by the Charadriiformes, Procellariiformes and Pelecaniformes (Table 1)”

47) *The lithology that the seabirds reside in would have a big impact on the net longer-term importance of the fate of excreted N and P (maybe particularly P). This is why the Pacific atolls were such rich phosphate rock resources—excreted P onto a pure carbonate substrate rapidly produces relatively insoluble carbonate fluorapatite. This is a net loss in terms of reactive P...until, that is, we mined it all up for fertilizers! Even a brief reference to the mineralogy of the underlying lithology would help to clarify the longer-term implications of the seabird excretion component of these cycles. Could be inserted in the discussion from line 242 to 248.*

We agree with the Reviewer. Seabird colonies act to some extent as sinks of N and, fundamentally, of oceanic P. These processes acquire special relevance in the guanera zones, but these areas are very few in the world because they require climatic conditions of extreme aridity. The rest of the areas where most of the colonies are located are those in which during some period of the year there is a water excess. In this way, important quantities of fecal material accumulated during the breeding season is lost to the marine or continental waters, which causes no guano accumulation (see also comment # 9, Reviewer 1). As shown in Figure 1 and in the Discussion section, the P is in an anionic form, so that in acid soils it can be adsorbed by soil colloids, or precipitate as calcium phosphate. This process must be important during the first years of existence of the colony, since the soil subsequently becomes saturated and stops adsorbing P (Otero et al., 2015). Perhaps the most relevant aspect of this mechanism is that it represents a slower release of P than that of N. In the new manuscript we have included additional data of the forms of N (NO_3^- and NH_4^+) and P that are lixibables (soluble in water or in neutral salts), representing the quantities of P and N that could be rapidly incorporated into the bodies of coastal or continental waters (e.g., rivers, lagoons). The obtained values can be considered as relevant, since we estimate that 12-21% of the total N and P reach relatively confined areas, or areas with poor water circulation or long renewal times, such as lakes, estuaries and inlets. These flows can play an especially important role in the primary production of these ecosystems and recent works appear to support this idea. For example, Lorrain et al. (2017) reported clear enrichments of N in the coastal waters of remote Pacific islets, which reached 100-400 m distance from the coast due to presence of seabird colonies. Additionally, Zhu et al. (2016), observed that the presence of penguin colonies are capable of modifying the current and ancient chemical and microbiological composition of water bodies.

On the other hand, recent work by our research group has already reported the mineralogy associated to seabird colonies in Antarctica (González-Guzman et al., 2016). This mineralogical analysis showed the presence of phosphates typical of latrine media, such as taranaquite [$\text{K}_3\text{Al}_5(\text{PO}_4)_2(\text{HPO}_4)_6 \cdot 18\text{H}_2\text{O}$], minyulite [$\text{KAl}_2(\text{OH},\text{F})(\text{PO}_4)_2 \cdot 4\text{H}_2\text{O}$], leucophosphate [$\text{KFe}_2(\text{PO}_4)_2(\text{OH}) \cdot 2\text{H}_2\text{O}$], struvite [$(\text{NH}_4)\text{Mg}(\text{PO}_4)_6 \cdot (\text{H}_2\text{O})$] and hydroxylapatite [$\text{Ca}_5(\text{PO}_4)_3\text{OH}$].

To address the Reviewer' suggestion, the following text was added to the new manuscript [lines 239-248]:

“Results show that 12.7% of the total excreted N corresponds to labile forms of this element, which can be readily lixiviated toward continental or coastal waters (Table 1). This process can become especially relevant in the Antarctic and Southern Ocean regions, where the majority of the main guano producers, represented by the Sphenisciforme population, is concentrated (Table 1).

In regard to labile P, one of its main differences relative to N is that the former lacks a stable gaseous phase (phosphine, or PH_3)⁴², making the concentrations of this compound in seabird excrements extremely low⁴³. Results show that the global emission of P to the atmosphere from faecal material produced by seabirds can be considered as negligible ($1.5 \times 10^{-6} \text{ Gg y}^{-1}$). On the other hand, mobilization of P via leaching can be restricted due to its adsorption to soil colloids⁶ (Fig. 1)."

And [lines 253-261 of the new manuscript]:

"However, the majority of the seabird colonies' soils are located on rocky substrates, with shallow and sandy soils or where the occurrence of permafrost near the soil surface has a strong regulatory effect on leaching and development processes⁴⁷⁻⁴⁸. Under these conditions, a rapid P soil saturation is eventually reached and this element is then lixiviated towards coastal or continental waters⁶. In this sense, the results obtained in this work show that 21% ($0.02 \times 10^{-3} \text{ Gg y}^{-1}$) of the total excreted P is readily lixiviated. Hence, the fraction of P excreted by seabirds can be considered as relevant relative to the total P flux to the oceans since it represents 2.5% and 10% of the total dissolved fluxes of inorganic and organic P by rivers, respectively (Supplementary Table 3)."

As well as [lines 336-344 of the new manuscript]:

"Under the general term of labile N and P species were considered those readily leachable forms which, in short time spans (e.g., months), can reach marine coastal or continental (lakes, rivers, etc.) waters. Reported average concentrations of nitrogen ($\text{NO}_3^- + \text{NH}_4^+$) soluble in water or in neutral salts (e.g., KCl, MgCl_2 1 M) that are present in faecal material were considered as labile N species^{6,45}. For the case of P, the concentration of phosphate soluble in the extract Mehlich 3 was also included within the labile fraction and considered as bioavailable⁶. All relevant information pertaining labile N and P concentrations were obtained from data reported in the literature (Supplementary Table 4), and from information obtained from analyses of the excrements of *Larus michaellis*⁶. Additionally, volatile P (as phosphine) was assumed to have an average value of $8.8 \times 10^{-6} \text{ mg kg}^{-1}$ of PH_3 in faecal materials⁴³."

See also Supplementary Table 3.

Finally, the following paragraph was added to the new manuscript [lines 177-180]:

"On the other hand, the amounts of excreted labile forms of N and P (those that can be readily dissolved) were 72.5 Gg y^{-1} and 21.8 Gg y^{-1} , respectively, with the highest values corresponding to the Sphenisciformes (35.0 Gg N y^{-1} and 10.9 Gg P y^{-1} , respectively), followed by the Charadriiformes, Procellariiformes and Pelecaniformes (Table 1)"

48) Consider replacing the term "New" in the current title with "Important." Seabird colonies are not new, but rather they have an important role in global nutrient cycles.

We agree with the Reviewer. The title was changed from:

"SEABIRD COLONIES AS NEW GLOBAL DRIVERS IN THE NITROGEN AND PHOSPHORUS CYCLES"

To:

"SEABIRD COLONIES AS IMPORTANT GLOBAL DRIVERS IN THE NITROGEN AND PHOSPHORUS CYCLES"

Trivial dits:

49) Line 32 add "respectively"

The text was corrected as suggested by the Reviewer.

50) Line 89 should be “excrement”

The text was corrected as suggested by the Reviewer.

51) Line 147 typo with Thickthick

The text was corrected as suggested by the Reviewer.

52) Line 182 change to “over tens and hundreds of million years”

The text was corrected as suggested by the Reviewer [line 190 of the new manuscript].

53) Line 474 should be “nutrient”

The text was corrected as suggested by the Reviewer.

54) Line 476 change to “waters it is...”

The text was corrected as suggested by the Reviewer.

55) Line 479 remove “that”

The text was corrected as suggested by the Reviewer.

56) Line 490 change to “simpler cycle”

The text was corrected as suggested by the Reviewer.

57) Line 491 replace “under” with “in”

The text was corrected as suggested by the Reviewer.

58) Line 492 although lixiviated is the proper term, perhaps also add parenthetically “solubilized”

The text was corrected as suggested by the Reviewer.

59) Table 1 has Gg N/yr under the Total P Excreted column heading

The text was corrected as suggested by the Reviewer.

**Gabriel Filippelli
IUPUI**

REVIEWERS' COMMENTS:

Reviewer #1 (Remarks to the Author):

SEABIRD COLONIES AS IMPORTANT GLOBAL DRIVERS IN THE NITROGEN AND PHOSPHORUS CYCLES

Huerta-Diaz et al.

The products of this paper are 1) an updated seabird population estimate and 2) the first global map of annual P excretion rates from seabirds 3) potential N and P flux estimates from land back into the ocean. The paper is well written and reviews previous seabird N work comprehensively. The work extends Riddick et al. (2012) by presenting P excretion rates.

The seabird population estimate is similar to values published in other papers. The penguin population estimate is lower than I would expect, this is worth double checking. If the authors are confident this is a result of population change, then this would be a very important finding. More effort could be spent identifying population changes and how this affects/updates N & P excretion rates what impact this would have on the surrounding ecosystem.

The results section is clear but some of the content should be moved to the Discussion. The Discussion is very long and contains content that would be better placed in an Introduction.

L50 – L56 Verbose and unnecessary.

L52 Replace “was” with “wasn’t”.

L61 Delete “are varied and”

L63 – L71 Are there any more recent studies with evidence that show the effect of seabirds on plant growth?

L74 Which studies in the Mediterranean, Atlantic etc?

L75 Explain what “local interest” means

L85 “ref-6”?

L88 Remove “updated”

L97 Individuals?

L98 Replace “, whereas” with "and"

L100 Again, individuals?

L113 replace “superior” with “larger”

L120 Is it worth mentioning nesting behaviour and climate? Burrow nesters will not affect above ground biogeochemistry. Penguins nesting on ice will have little effect.

L187-209 Move to Introduction, this isn't Discussion.

L210 What “Results”?

L244 What results?

L264 – 266 This sentence doesn't make sense.

L266 What is the implication of this?

S. Riddick

Reviewer #2 (Remarks to the Author):

I was happy to read your rebuttal and revisions to your initial manuscript. It makes sense about the problems with the ordering of the file and population size, which was my significant concern in the first MS version. This version makes much more sense and you have explained well the sections that I felt needed a bit more clarity.

My very minor suggestions are:

L119 – kg not Kg

L240 – bird life, not birdlife

L247 – Sphenisciform

I would have to defer to other reviews on any changes on the chemical modelling etc.

Reviewer #3 (Remarks to the Author):

The authors have extensively revised their manuscript in response to my review comments, as well as those of the other reviewers. I have no further comments to make based on my assessment, which largely revolved around the importance of this contribution wrt global phosphorus cycling.

RESPONSE TO REVIEWERS

Reviewer 1

1) L50 – L56 Verbose and unnecessary.

We partially agree with the Reviewer and we decided to eliminate the paragraph (L52-L56 of the old manuscript):

“However, it was until the second half of the 19th century that European researchers identified Peruvian guano as one of the richest sources of N ever discovered⁸. Thus, guano from seabird colonies fueled the great demand for food that happened in the northern hemisphere, promoting the development and growth of the European population.”

However, we decided to keep L50-L52 of the old manuscript because we considered this part relevant in the context of the article.

2) L52 Replace “was” with “wasn’t”.

Since L52-L56 were eliminated from the manuscript (see comment 1, above), it was not necessary to incorporate this change into the new manuscript.

3) L61 Delete “are varied and”.

The change was made according to the Reviewer’s suggestion.

4) L63 – L71 Are there any more recent studies with evidence that show the effect of seabirds on plant growth?

In this paragraph there are recent references that were already cited; for example, Reference 13 (Lorrain et al., 2017) and 6 (Otero et al., 2015). However, in this part of the manuscript a special emphasis was made to the oldest citations for the purpose of proposing a historical approach on the study of the effects of seabirds on the environment.

5) L74 Which studies in the Mediterranean, Atlantic etc?

To address the Reviewer’s observation, the sentence was changed from (L72-L76 of the old manuscript):

“Although in some cases seabird colonies have profoundly altered the biogeochemical processes that occur in coastal surface systems (soils, sediments, waters) and have transformed plant communities (for example, Mediterranean and Atlantic islands, North-East Scottish coast, Pacific reef corals), most studies revealing biogeochemical and ecological alterations have been mostly of local interest^{6,9-14} (Fig. 1).”

To (L86-L90 of the new manuscript):

“Although in some cases seabird colonies have profoundly altered the biogeochemical processes that occur in coastal surface systems (soils, sediments, waters) and have transformed plant communities (for example, Mediterranean and Atlantic islands^{6,9}, North-East Scottish coast¹¹, Pacific reef corals¹³), most studies revealing biogeochemical and ecological alterations have been mostly of local interest and importance to particular areas^{6,9-14} (Fig. 1).”

6) L75 Explain what “local interest” means

To address the Reviewer’s observation, the text was changed from (L76-L78 of old manuscript):

“...most studies revealing biogeochemical and ecological alterations have been mostly of local interest”

To (L89-L90 of new manuscript):

“...most studies revealing biogeochemical and ecological alterations have been mostly of local interest and importance to particular areas”

7) L85 “ref-6”?

The reference is correct: The article cited as reference 6 is a review article dealing with P concentrations in marine seabird excrements at the worldwide level.

8) L88 Remove “updated”

We substituted “updated” by “obtained from”. The text in the new manuscript now reads as (L111-L112):

“For this purpose, current estimates of the world seabird populations were obtained from global seabird population data published by international organizations.”

9) L97 Individuals?; L98 Replace “, whereas” with "and"; L100 Again, individuals?.

The paragraph from the old manuscript was changed from (L97-L100):

“The worldwide population of breeding seabirds and chicks were estimated to be 804 million (Table 1, Supplementary Table 1), whereas the total population, including 30% of non breeding seabirds, was estimated to be 1045 million. Similar results have been obtained in previous studies, with total population estimates ranging from 900 to 1180 million^{18,22}.”

To (L124-L126):

“The worldwide population of breeding seabirds and chicks is estimated to be 804 million individuals (Table 1, Supplementary Data 1), and the total population, including 30% of non breeding seabirds, is estimated to be 1045 million individuals.”

10) L113 replace “superior” with “larger”

The change was made according to the Reviewer’s suggestion.

11) L120 Is it worth mentioning nesting behaviour and climate? Burrow nesters will not affect above ground biogeochemistry. Penguins nesting on ice will have little effect.

We think this comment is partially true. However, the most important fact is that seabirds concentrate high quantities of N and P in the area where they place their breeding colonies. Furthermore, the relevance of ornithotrophication is that it is limited not only to the breeding area, but also to surrounding continental and marine waters, since nutrients can be lixiviated to these waters, independently of the nesting behavior (ice, burrows). On the other hand, as it is mentioned in the Discussion section, soils from seabird colonies undergo important geochemical and mineralogical changes, enriching themselves in the subsurface layers in N and P, which become also bioavailable for plants. Hence, plants can use not only the N and P deposited on the surface, but also the ones concentrated in the subsoil.

12) L187-209 Move to Introduction, this isn’t Discussion.

We disagree with the Reviewer. We think that these two paragraphs are important to the general reader because it can be useful to frame the importance and environmental relevance of seabird colonies.

13) L210 What “Results”?

To address the Reviewer’s observation, we changed the text in the old manuscript from (L210-L211):

“Results show that most of the compartments of the N cycle contain between three and six times more N than that excreted by seabirds in the breeding colonies (Supplementary Table 2).”

To (L261-L262 of the new manuscript):

“The global biogeochemical cycle of N shows that most of its compartments contain between three and six times more N than that excreted by seabirds in the breeding colonies (Supplementary Table 1).”

14) L264 – 266 This sentence doesn't make sense.

To address the Reviewer's observation, the sentence was changed from (L264-L266 of the old manuscript):

"Hence, the fraction of P excreted by seabirds can be considered as relevant relative to the total P flux to the oceans since it represents 2.5% and 10% of the total dissolved fluxes of inorganic and organic P by rivers, respectively (Supplementary Table 3)."

To (L330-L332 of the new manuscript):

"This labile phosphorus flux can be considered as important, since it represents 2.5% and 10% of the total dissolved fluvial fluxes of inorganic and organic P, respectively (Supplementary Table 3)."

15) L266 What is the implication of this?

We think that the implications of this and our other findings are clearly described in the next three paragraphs (see L333-L362 of the new manuscript) and, implicitly, in the values that the labile P represent with respect to the fluvial P fluxes.

Reviewer 2

16) L119 – kg not Kg.

The change was made according to the Reviewer's suggestion.

17) L240 – bird life, not birdlife.

The change was made according to the Reviewer's suggestion.

18) L247 – Sphenisciform.

The change was made according to the Reviewer's suggestion.